# Optimizing Diversity and Quality through Base-Aligned Model Collaboration

Yichen Wang [* 1]   Chenghao Yang [* 1]   Tenghao Huang [* 2]   Muhao Chen [3]   Jonathan May [2]   Mina Lee [1]

## Abstract

Alignment has greatly improved large language models (LLMs)' output *quality* at the cost of *diversity*, yielding highly similar outputs across generations, especially in open-ended generation tasks. We propose Base-Aligned Model Collaboration (BACO), an inference-time token-level model collaboration framework that dynamically combines a base LLM with its aligned counterpart to optimize diversity and quality. Using uncertainty and content-based signals, BACO employs *routing strategies* to determine, at each token, which model to decode from. Prior diversity-promoting methods often improve diversity at the expense of quality or require expensive decoding or post-training. In contrast, BACO achieves both high diversity and quality post hoc within a single pass, while offering strong controllability. We introduce a family of effective routing strategies and evaluate them across three open-ended generation tasks with 13 diversity and quality metrics. BACO consistently surpasses state-of-the-art inference-time baselines. With our best router, BACO achieves a 21.3% joint improvement in diversity and quality, which is further supported by human evaluations. Overall, our results demonstrate that collaboration between base and aligned models provides an effective and controllable mechanism for optimizing the diversity-quality trade-off. Code, data, and demo are available at https://yichenzw.com/base-aligned-collab.github.io/.

## 1. Introduction

While alignment greatly improved large language models (LLMs)' output *quality* in terms of instruction following

and downstream task performance (Ouyang et al., 2022), it results in a stark reduction in output *diversity* (Kirk et al., 2023; Zhang et al., 2025; West & Potts, 2025; Spangher et al., 2025b; Huang et al., 2025). Across repeated sampling, a model after alignment (i.e., aligned model) tends to produce highly similar outputs, whereas a model before alignment (i.e., base model) yields diverse outputs. For example, when prompted with "suggest a summer trip destination in the US," a base model may produce diverse destinations across generations, while the aligned model often converges on a single dominant one (Figure 1A). This *diversity-quality trade-off* undermines utility in open-ended generation tasks (e.g., creative writing and dialogue) by encouraging formulaic language use (Zhang et al., 2024; Chakrabarty et al., 2025), diminishing creativity (West & Potts, 2025), and suppressing ideation in human-AI interaction (Padmakumar & He, 2023; Meincke et al., 2025; Ashkinaze et al., 2025; Spangher et al., 2025a;c). These findings motivate methods to improve diversity in aligned LLMs.

Prior diversity-promoting methods attempt to address the diversity-quality trade-off at both the training and inference stages (§A). The former (Lanchantin et al., 2025; Chung et al., 2025; Li et al., 2025) incorporate explicit diversity objectives into preference optimization during reinforcement learning. While effective at improving diversity, such methods require modifying the model's output distribution, which can compromise desirable alignment properties such as safety and helpfulness (Qi et al., 2023). The latter consists of decoding-based techniques, such as adjusting temperature and beam search (Vijayakumar et al., 2016), as well as prompt-based techniques, including in-context learning (Meyerson et al., 2024), prompt paraphrasing (Zhang et al., 2025), and multilingual back-translation (Wang et al., 2025b). However, these inference-time techniques typically require multiple decoding passes or long-horizon planning to improve diversity, and may still disproportionately degrade generation quality (Peeperkorn et al., 2024). This reveals a fundamental limitation of forcing a single model to excel at both diversity and quality.

To overcome this limitation, we introduce Base-Aligned Model Collaboration (BACO), an inference-time token-level model collaboration framework (§3) that combines the complementary strengths of a base model for diversity and its aligned counterpart for quality. BACO operates

---
[*]Equal contribution   [1]University of Chicago [2]University of Southern California, Information Sciences Institute [3]University of California, Davis.   Correspondence to: Yichen Wang <yichenzw@uchicago.edu>.

*Proceedings of the 43rd International Conference on Machine Learning*, Seoul, South Korea. PMLR 306, 2026. Copyright 2026 by the author(s).

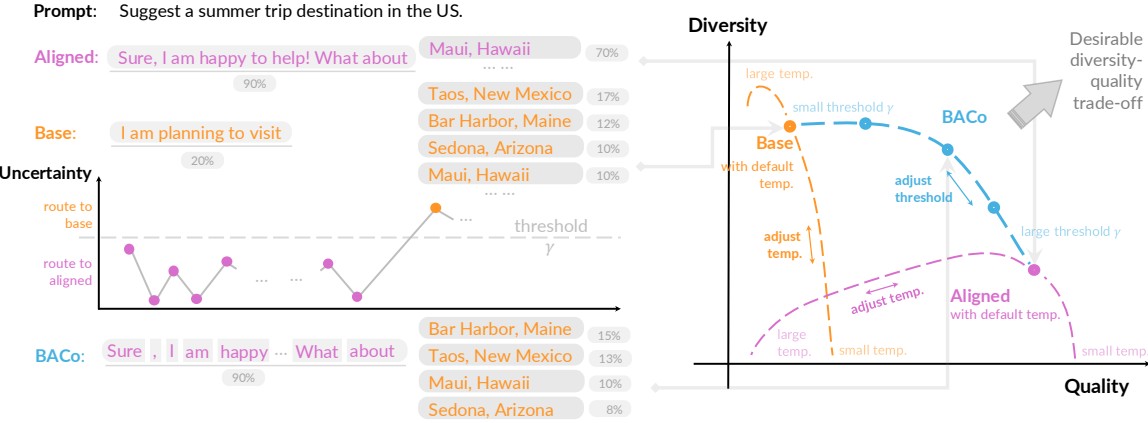

(A) Example outputs from base, aligned, and BACO models. (B) The diversity-quality trade-off space.

*Figure 1.* BACO is an inference-time token-level model collaboration framework that combines a base model's diversity with its aligned counterpart's quality. (A) A comparison of generated outputs. The aligned model produces high-quality but low-diversity outputs, while the base model produces high-diversity but low-quality outputs. BACO optimizes both diversity and quality by dynamically *routing* between them. The probabilities of token(s) are in grey next to text boxes. (B) Illustration of the diversity-quality trade-off space. Single models face a steep trade-off, where improving diversity by adjusting configuration (e.g., by increasing temperature) degrades quality. BACO achieves a better Pareto curve and allows for easy traversal across this frontier by adjusting the router's threshold.

via a lightweight, token-level *routing strategy* that dynamically switches between the two models in a single decoding pass, requiring no fine-tuning or prompting (Figure 1A). This design is motivated by recent findings suggesting that collaboration between base and aligned models is feasible. Specifically, Fei et al. (2025) demonstrate that base and aligned models largely agree on next-token predictions, a phenomenon known as *superficial alignment* (Lin et al., 2023). In contrast to the prior work that leverages this phenomenon to improve base-model quality, our goal is to jointly optimize diversity and quality, surpassing either model alone through collaboration (Figure 1B).

Importantly, the diversity-quality trade-off cannot be resolved by a fixed design. Different tasks naturally favor different operating points along this spectrum, and user preferences may further vary based on context. A practical framework, therefore, requires *controllability*, the ability to adjust generation along the diversity-quality spectrum on demand. BACO provides such controllability through an adjustable routing threshold (Figure 1), which continuously tunes the contribution of the base and aligned models during decoding. In addition, we introduce a family of routing strategies within a unified design space (§3.1) that consistently improve the diversity-quality trade-off. These strategies capture complementary signals and exhibit different strengths across diversity dimensions, providing an additional axis of control beyond the routing threshold.

We validate our approach across three open-ended generation tasks: instruction following, dialogue, and creative writing (§4). In open-ended generation, both diversity and quality can be defined in numerous ways, and no single met-ric provides a complete evaluation. Therefore, rather than treating evaluation as a single objective, we formulate it as a collection of bi-dimensional diversity-quality trade-off spaces, each defined by a specific pair of metrics. Focusing on diversity, we evaluate 11 diversity and 2 quality metrics, resulting in $11 \times 2$ spaces, to assess BACO's consistent improvements. In addition to instruction-following and dialogue tasks which have relatively short outputs, we also evaluate long-form generation, where we measure long-form diversity at the discourse level in terms of plot structure and emotional flow (Tian et al., 2024). Our evaluation focuses on open-ended generation settings where diversity is a first-class objective, for which benchmarks designed primarily for single-answer accuracy (e.g., math and code reasoning) are less suitable. Nevertheless, as a cross-check, we additionally evaluate BACO on tasks with verifiable quality criteria, including mathematical reasoning and verifiable instruction following, to ensure that diversity improvements do not come at the cost of severe quality degradation. We complement automatic evaluations with human evaluations of diversity and quality.

In our experiments (§5), we demonstrate that BACO achieves a new state-of-the-art in optimizing the diversity-quality trade-off in inference time. Overall, BACO with our best router achieves a 21.3% improvement over the strongest inference-time baselines, with even larger gains on semantic diversity metrics. These improvements are consistent across tasks and evaluation metrics, and are further supported by human evaluations. By viewing the base and aligned models as two checkpoints during training, our results suggest that collaboration across checkpoints can enable effective

control over diversity and quality. Overall, BACo provides a simple framework for base-aligned model collaboration, effectively improving both diversity and quality.

In summary, our contributions are threefold:

① We propose BACo, an inference-time token-level model collaboration framework that combines a base model and its aligned counterpart, along with a family of lightweight routing strategies, to produce high-diversity and high-quality outputs across generations.

② We formulate the diversity-quality trade-off as a collection of bi-dimensional evaluation spaces and conduct a comprehensive evaluation across $11 \times 2$ metric pairs, including long-form diversity and human evaluation.

③ Through extensive experiments on three open-ended generation tasks (i.e., instruction following, dialogue, and creative writing), we show that BACo consistently outperforms strong baselines.

## 2. Preliminary

**Large Language Models (LLMs).** LLMs are typically trained to autoregressively predict the next token of the output $y$ given a prompt $x$. The conditional probability is factorized as $P(y|x; \theta) = \Pi_t P(y_t|[x, y_{<t}]; \theta)$, where $y_{<t}$ denotes the output prefix generated up to position $t$-1, and $\theta$ denotes the model parameter.

Alignment is the process of fine-tuning an LLM to align its outputs with human intent, ethical principles, and desired behavioral norms, typically through instruction tuning or reinforcement learning from human feedback (RLHF) (Ouyang et al., 2022; Bai et al., 2022). We use *base models* to denote models without alignment tuning (e.g., Llama-3-8B) and *aligned models* to denote those further optimized with alignment (e.g., Llama-3-8B-Instruct) (Dubey et al., 2024).

**Diversity and Quality Measurement.** In this paper, we measure diversity over a group of outputs independently generated from the same prompt $x$: $\mathcal{Y}(x) = \{y^{(1)}, \dots, y^{(k)}\}, y^{(i)} \sim P(\cdot \mid x; \theta)$. This group-level diversity is denoted as $D(\mathcal{Y})$ (Kirk et al., 2023; West & Potts, 2025) (e.g., the clustering-based approach in Kuhn et al. (2023)). Quality is modeled as $Q(y|x)$ for each output given the prompt, typically by a reward model or human evaluator, reflecting linguistic fluency and instruction-following (Lambert et al., 2024; Zhang et al., 2025). The group-level quality is then defined as the average quality across all outputs in a group: $Q(\mathcal{Y}) = \sum_{i=1}^{k} Q(y^{(i)}|x)$. For simplicity, we refer to group-level diversity and group-level quality as diversity and quality in this work.

| Model | Diversity (#Clusters) ↑ | Quality (Reward) ↑ |
|---|---|---|
| Llama-3-8B | **8.13** | 1.28 |
| Llama-3-8B-Instruct | 2.58 | **7.62** |

*Table 1.* Diversity and quality of Llama-3-8B's base and aligned models in our preliminary experiment. The results demonstrate a clear diversity-quality trade-off in the two model's performance. Diversity is measured by the number of semantic equivalent clusters of the output group, and quality is the average reward per output from another LLM.

**Diversity-Quality Trade-off.** Alignment, while effective at improving output quality, comes at the cost of reduced output diversity (Lu et al., 2025; West & Potts, 2025; Yang & Holtzman, 2025). To demonstrate this trade-off and quantify its magnitude, we run a preliminary experiment with Llama-3 on a subset of WildChat (Zhao et al., 2024). We first situate the performance of the base and aligned models with default configuration (the two noted points in Figure 1B) within the diversity-quality space. Following the evaluation protocol of the diversity-focused benchmark NoveltyBench (Zhang et al., 2025), we evaluate Llama-3's base and aligned models on an open-ended subset of Wild-Chat (Zhao et al., 2024). Here, diversity is measured as the number of semantic equivalent classes of the output group via Zhang et al. (2025)'s clustering, and quality is measured as the average reward per output from Skywork-Reward-Gemma-2-27B (Liu et al., 2024).[1] We sample 10 outputs per prompt. As illustrated in Table 1, the diversity-quality trade-off is stark: the base model is **3.15x more diverse**, whereas the aligned model has **5.95x higher quality**. Inherently, this performance trade-off stems from alignment's tendency to reduce the entropy of the next-token prediction distribution, concentrating probability mass on fewer, high-quality tokens, a phenomenon known as *mode collapse* (Lin et al., 2023; Shumailov et al., 2024; Hamilton, 2024; Yang & Holtzman, 2025; Cui et al., 2025).

This presents a dilemma: one can either use a high-diversity but low-quality base model, or a high-quality but low-diversity aligned model. The single-model paradigm is insufficient, as neither extreme is ideal for all applications. Hence, we argue that an ideal method is able to pursue the best of both worlds. To this end, we formalize the problem in a two-dimensional *diversity-quality space* $S = \{(D, Q)\}$. In this space, any given method under specific configurations $\gamma$ (e.g., sampling parameters) is evaluated to be a single point. An ideal method, by adjusting $\gamma$, should approximate the *Pareto frontier*: the set of optimal solutions where diver-

---

[1]These are two of many possible measurements introduced later in the paper. We use them here as representative examples for the pilot study, as they are among the widely adopted metrics.

sity cannot be improved without sacrificing quality, and vice versa. This frontier represents the best possible trade-offs.

## 3. BACO: Base-Aligned Collaboration for Diversity and Quality

Recent work has provided empirical evidence for the superficial alignment hypothesis (Zhou et al., 2023a; Lin et al., 2023), which suggests that a base model and its aligned counterpart largely agree on next-token predictions. Building on this observation, Fei et al. (2025) show that introducing only a small fraction of aligned-model tokens into a base model's decoding can recover task-specific performance comparable to that of the aligned model. However, such approaches primarily focus on improving the quality of the base model, without explicitly addressing the loss of diversity introduced by alignment.

Motivated by this gap, we hypothesize that

> Collaboration between a less-aligned, higher-diversity model and a more-aligned, higher-quality model during inference can better balance the diversity-quality trade-off than either model alone.

Base and aligned models form a natural pair to test this hypothesis, as they are readily available off the shelf and exhibit complementary strengths.

Based on this hypothesis, we propose **BACO**, an inference-time framework that orchestrates collaboration between a base model ($P_{\text{base}}$), serving as a source of diversity, and its aligned counterpart ($P_{\text{aligned}}$), serving as a source of quality, at the token level.[2]

At the core of BACO is a **router**, a lightweight decision module that determines, at each decoding step, which model should generate the next token. The router operates according to **routing strategy(ies)** $\mathcal{R}$, which selects between base and aligned models accordingly.[3] Intuitively, the router acts as a "gatekeeper": it routes the next token generation to the base model when diversity is desired and to the aligned model when quality is desired.

Formally, BACO orchestrates the two models as:

$$P_{\text{BACO}}(y_t|c_t) = w_{\text{base}} \cdot P_{\text{base}}(y_t|c_t; \theta_{\text{base}}) \\ + (1 - w_{\text{base}}) \cdot P_{\text{aligned}}(y_t|c_t; \theta_{\text{aligned}}) \quad (1)$$

where $c_t = [x, y_{<t}]$, and the gating weight $w_{\text{base}} \in \{0, 1\}$ for each candidate token $y_t$ is given by the router:

$$w_{\text{base}} = \mathbb{I}\left[\mathcal{R}\left(y_t|c_t, P_{\text{base}}, P_{\text{aligned}}\right) = \text{base}\right] \quad (2)$$

---

[2]The token-by-token nature of LLM autoregressive decoding makes token-level control feasible.

[3]The router can be based on a single routing strategy or combine multiple ones.

In practice, since one word may consist of multiple tokens, we restrict switching to word boundaries to prevent erroneous generation when the two models use different tokenizations. Full decoding pseudocode is provided as Algorithm 1 in Appendix B; further implementation and discussion are in Appendix B and Appendix I.

### 3.1. Routing Strategy Design

At the core, BACO is a lightweight router that determines which model to route to at each decoding step. A routing strategy specifies (*i*) what information to use as prior for routing decisions, and (*ii*) which model to switch to given the information. Each strategy includes a *threshold parameter* $\gamma$ that, analogously to decoding temperature, continuously adjusts the diversity-quality balance: larger $\gamma$ biases toward the base model (more diversity), smaller $\gamma$ biases toward the aligned model (more quality).

We design routing strategies based on two complementary signal categories: logit-based and content-based. Each category captures a distinct yet widely applicable perspective.

**Logit-Based.** Logit-based strategies leverage the next-token prediction distribution to infer the model uncertainty, reflecting a *model-centric* perspective (Fei et al., 2025; Zheng et al., 2024; Leviathan et al., 2023a). High uncertainty, indicated by low maximum probability or high entropy, suggests that multiple continuations are plausible, making such positions natural opportunities for diversification (Yang & Holtzman, 2025; Wang et al., 2025c). Concretely, we implement: ① **BACO-P** routes to the base model when its maximum token probability falls below a threshold $\gamma$, i.e., $\max_{y_t} P_{\text{base}}(y_t \mid \cdot) < \gamma$; ② **BACO-H** routes to the base model when its next-token entropy exceeds, i.e., $H_{\text{base}}(Y_t \mid \cdot) = \sum_{y_t} P_{\text{base}}(y_t|\cdot) \log P_{\text{base}}(y_t|\cdot) > \gamma$; etc.

**Content-Based.** Content-based strategies adopt a *language-centric* perspective, making routing decisions based on the semantic roles of predicted tokens. The motivation is twofold. First, linguistic structures such as content words often correspond to semantic or stylistic branch points where diversity is most perceptible to humans (Yao et al., 2019; Sims et al., 2019). Second, Lin et al. (2023); Fei et al. (2025) suggest that disagreements between base and aligned models often arise over stylistic tokens, such as formatting tokens (e.g., '\n') or function words (e.g., 'and', 'if'). We therefore implement: ① **BACO-PUNC** routes to the aligned model when its top-ranked token is either a punctuation or formatting token; ② **BACO-FC** routes to the aligned model for function words to preserve stylistic coherence. Content-based strategies are also applicable to black-box models, as they do not require access to logits.

In practice, combining logit-based and content-based strategies yields the strongest performance, as they rely com-

plementary signals. For example, BACO-P-FC prioritizes function-word routing (-FC) before falling back to probability-based decisions (-P). In the next section, we evaluate a wide range of routing strategies and find that many variants are effective. For clarity, we present representative strategies in the main paper, with additional variants and implementation details in Appendix B.2. We focus on the two categories of routing strategies and leave other routers (e.g., learned routers) for future work.[4]

## 4. Experimental Setup

We design our experiments to empirically validate BACO's central goal: to optimize the diversity-quality trade-off.

**Datasets.** We mainly evaluate BACO across three representative open-ended generation tasks: NoveltyBench (Zhang et al., 2025) for instruction following, WildChat (Zhao et al., 2024) for dialogue, and Narrative-Discourse (Tian et al., 2024) for creative writing. Together, these datasets cover both short- and long-form open-ended generation across varying levels of task complexity. For full dataset details, we refer readers to Appendix C.

**Baselines.** We compare BACO with inference-time methods across five categories: ① Single-model: a base model or an aligned model, each sampled at varying temperatures. ② Prompting-based: in-context resampling (Meyerson et al., 2024; Zhang et al., 2025), where $n$ outputs are generated sequentially within a single dialogue; paraphrase prompting (Jiang et al., 2020; Zhang et al., 2025), where paraphrased variants of the same instruction are used to increase output diversity. ③ Decoding-based: Diverse Beam Search (Vijayakumar et al., 2016), where a diversity penalty was added to the beam search algorithm. ④ Ensemble-based: response ensemble, where $n/2$ outputs are sampled from the base model and its aligned counterpart, and pooled into a single group; logit ensemble, which merges the next-token probability distributions of the two models before sampling. ⑤ Collaboration-based: NUDGING (Fei et al., 2025), where an aligned model selectively introduces tokens during a base model's decoding. Note that diverse beam search, paraphrase prompting, and in-context resampling require additional computation; in-context resampling does not perform parallel sampling.[5] These methods therefore provide more competitive reference points. The inference setups and experimental scope are provided in Appendix E.

---

[4]We exclude learned routers because (1) simple heuristics already yield substantial gains; (2) diversity is inherently multidimensional and cannot be captured by a single metric, leading to conflicting objectives and unstable training when jointly optimized (§4.1); and (3) diversity evaluation requires group-level sampling, which would substantially increase training cost and complexity.

[5]A wall-clock runtime comparison across all inference-time methods is provided in Appendix B.

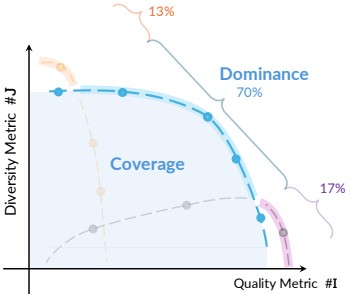

*Figure 2.* Illustration of the indicators on diversity-quality space: **Coverage**, the area under a method's trade-off curve (blue shading for the blue method); **Dominance**, the proportion of the global Pareto frontier (highlighted curves) contributed by the method.

**BACO.** Our experiments leverage two open-weight model pairs: `Llama-3-8B` and `Llama-3-8B-Instruct` (Grattafiori et al., 2024), and `Olmo2-7B` and `Olmo2-7B-Instruct` (OLMo et al., 2024) as they are widely used in literature (e.g., Fei et al. (2025)). We implement the single-strategy routers (e.g., -P which is based on maximum token probability) from Section 3.1 and denote multi-strategy routers as "-X-Y", where strategy Y precedes X (e.g., -P-PUNC, -P-FC, and -H-PUNC). These implementations serve as representative examples that demonstrate the possible design space of the BACO framework. BACO framework works well with a wide range of routers. We include two basic routers as baselines: ① -RAND routes to the base model by random chance $\gamma$; ② -JUDGE employs an external model to evaluate candidate tokens and makes a routing decision. Refer to Appendix B.2 for more details.

### 4.1. Automatic Evaluations

We next evaluate the diversity-quality trade-off of each method using automatic metrics. Prior work has proposed a wide range of diversity evaluation that apply different lexical and semantic metrics, reflecting different perspectives on language diversity. Since our goal is to improve general diversity and quality rather than optimize for any specific metric, we adopt 11 established diversity metrics and 2 quality metrics which form $11 \times 2$ diversity-quality spaces, and then aggregate their results.

Moreover, we aim to quantify the *controllability* of each method, i.e., the ability to adjust along the diversity-quality spectrum according to task or preference. As shown earlier in Figure 1B, each method is not evaluated as an individual point in a fixed configuration, but a curve formed by a sequence of points, illustrating the diversity and quality performance in different configurations. Each curve illustrates the trade-off of a specific method. To enable clear comparison, we apply two indicators from multi-objective optimization, `Coverage` (Cov.) and `Dominance` (Dom.), to aggregate the curve-shaped performance across all spaces

| Method | Lexical | | Semantic | | Overall | |
|---|---|---|---|---|---|---|
| | *Cov.* | *Dom.* | *Cov.* | *Dom.* | *Cov.* | *Dom.* |
| Base | 0.098 | 12.7% | 0.098 | 16.0% | 0.098 | 14.3% |
| Aligned | 0.269 | **49.0%** | 0.104 | 29.2% | 0.186 | **39.0%** |
| Nudging | 0.276 | 9.3% | 0.247 | 9.9% | 0.261 | 9.6% |
| Decoding | - | 0.3% | - | 0.3% | - | 0.3% |
| Prompting (Best) | - | 2.7% | - | 2.2% | - | 2.4% |
| Ensemble (Best) | - | 1.1% | - | 1.9% | - | 1.5% |
| BACo (Best) | **0.445** | 24.9% | **0.360** | **40.5%** | **0.403** | 32.7% |

*Table 2.* Averaged performance of all methods across all datasets and diversity–quality spaces. BACo consistently outperforms baselines across all semantic and most lexical spaces, demonstrating stronger controllability and substantially improving the semantic diversity–quality trade-off. The overall gains, as driven primarily by improvements in semantic, suggest that BACo produces more meaningful and content-level diversity, rather than superficial word-level changes, compared to other methods. See full results at Appendix G.

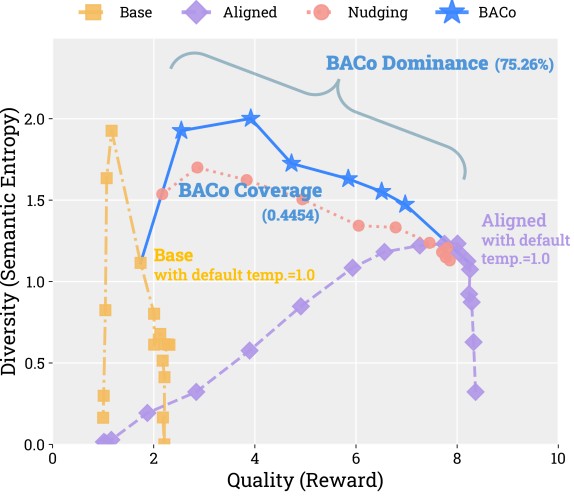

*Figure 3.* BACo's performance on one diversity-quality space (x: quality in terms of reward; y: diversity in terms of semantic entropy). Comparing with baselines, BACo (blue curve) attains larger Coverage of the top-right region and contributes to most of the Dominance, indicating improvement and controllability on diversity-quality trade-off.

into numerical results (Figure 2).[6]

**Coverage (Cov.)** indicator quantifies the area under a method's diversity-quality trade-off curve, following the hypervolume formulation used in multi-objective optimization. It measures how effectively a method traverses the diversity-quality spectrum as its control configuration varies. A larger Coverage value indicates that the method maintains good performance across a wide range of regions on diversity-quality spaces, offering usability to more different tasks or preferences overall.

**Dominance (Dom.)** indicator captures comparative optimality: whether and how often a method contributes to the global Pareto frontier among all methods. We compute the global Pareto frontier across all methods and apply the C-metric (Zitzler, 1999) to measure the portion of the frontier attributed to each method. A higher Dominance value indicates that a method achieves uniquely strong trade-offs unattainable by others.

These indicators are instantiated under the $11 \times 2$ diversity-quality spaces. And we report the average across the spaces.

**Lexical diversity** spaces use diversity metrics such as Distinct-$n$ (Dist-$n$; Li et al., 2015), EAD-$n$ (Liu et al., 2022), and Self-BLEU (Montahaei et al., 2019). **Semantic diversity** spaces rely on diversity metrics such as embedding cosine dissimilarity (Kirk et al., 2023), Vendi Score (embedding) (Friedman & Dieng, 2022), NLI diversity (Stasaski &

Hearst, 2022), and Semantic Entropy (Kuhn et al., 2023).[7] Since lexical and semantic metrics capture fundamentally different aspects of diversity, we analyze them separately in addition to reporting aggregated results. Increases in lexical diversity are relatively easy to achieve (for example, by raising the temperature), yet they mostly alter surface-level phrasing without changing meaning. In contrast, semantic diversity reflects deeper diversity in meaning, intent, and ideas, which is harder to elicit but more human-like and valuable in open-ended generation. **Quality** metrics include (*i*) perplexity under the aligned model, which reflects fluency and instruction following, and (*ii*) reward modeling scores predicted by `Skywork-Reward-Gemma-2-27B` (Liu et al., 2024), the state-of-the-art model on RewardBench (Lambert et al., 2024). These metrics are paired with diversity metrics to form subspaces. Finally, **overall** results average across all subspaces, yielding holistic method-level indicators. Hereafter, we use *lexical* to denote average results on all lexical diversity-quality spaces, *semantic* denotes semantic diversity-quality spaces, and *overall* averages all spaces in every result table. Full derivations and implementation details are in Appendix D.1.

---

[7]Each metric could include multiple variants. For example, Dist-1/2/3 for different n-gram, or cosine dissimilarity under different pretrained encoders. Each variance leads to a separate diversity-quality space.

---

[6]All curves in Figure 2 are illustrative only and do not correspond to the actual performance of any method.

| Routers | Lexical | | Semantic | | Overall | |
|---|---|---|---|---|---|---|
| | *Cov.* | *Dom.* | *Cov.* | *Dom.* | *Cov.* | *Dom.* |
| -RAND | 0.493 | 26.3% | 0.409 | 17.0% | 0.451 | 21.7% |
| -JUDGE | 0.302 | 2.6% | 0.254 | 0.6% | 0.278 | 1.6% |
| -P | 0.433 | 4.8% | 0.397 | 8.5% | 0.415 | 6.7% |
| -FC | 0.419 | 3.2% | 0.382 | 4.7% | 0.401 | 4.0% |
| -P-PUNC | **0.495** | **30.7%** | **0.452** | **31.3%** | **0.474** | **31.0%** |
| -H-PUNC | 0.466 | 16.4% | 0.427 | 18.6% | 0.446 | 17.5% |
| -P-FC | 0.435 | 16.0% | 0.406 | 19.2% | 0.421 | 17.6% |

*Table 3.* Averaged performance of routers within BACO on NoveltyBench across all diversity–quality spaces. The -P-PUNC router achieves the best overall performance. While the random router (-RAND) attains moderately strong results, mainly from increased surface-level lexical diversity, its performance drops sharply on semantic metrics, confirming that unguided switching fails to produce meaningful diversity. In contrast, -P-PUNC delivers the most balanced and consistent results across both lexical and semantic evaluations, showing combination of designed routing strategies leads to more meaningful diversity.

## 5. Results

We first compare BACO against strong inference-time baselines across datasets and diversity-quality spaces (§5.1). We then analyze the behavior of different routing strategies in a controlled setting (§5.2), followed by ablations on model pairing (§5.3). Finally, we evaluate long-form diversity (§5.4) and validate automatic metrics with human judgments (§5.5).

### 5.1. Overall Performance

Table 2 summarizes the performance of BACO and all baselines, aggregated across datasets and metrics. Overall, BACO improves Coverage by **0.142** and achieves **32.7%** Dominance across all evaluation spaces. Specifically, a Coverage improvement of 0.142 expands the achievable diversity-quality area by over 30% relative to the strongest baseline. And a Dominance of 32.7% indicates that BACO contributes nearly one-third of the global Pareto frontier over all baselines. The advantage is particularly pronounced in semantic diversity, where Dominance reaches **40.5%** (see Figure 3).[8] On the NoveltyBench dataset, the gap further widens, with Coverage improving by **0.274** and Dominance reaching **39.9%**. These trends are consistent across datasets and extend to the Olmo2 model family; full results are provided in Appendix §G. Qualitative output comparisons are shown in Appendix §K.

---

[8]The aligned model's high lexical dominance arises from its high sampling temperature, which produces long, low-quality sequences that artificially inflate diversity scores while reducing controllability.

**Validation beyond open-ended evaluation metrics.** To verify that these gains are not an artifact of open-ended evaluation metrics, we additionally evaluate BACO on two verifiable benchmarks: instruction following (IFEval) and mathematical reasoning (GSM8K). Across both tasks, BACO achieves higher diversity at matched quality or accuracy than the aligned baseline, confirming that the observed improvements are not only confined to open-ended generation.

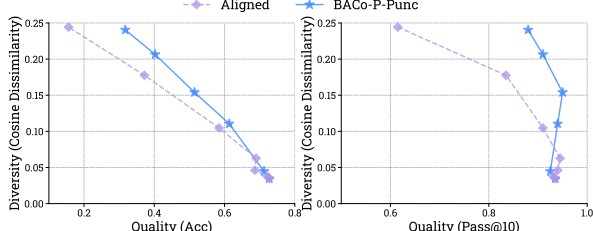

*Figure 4.* Diversity-accuracy trade-off comparison on GSM8K. At comparable accuracy levels, BACO achieves higher diversity than the aligned baseline, indicating that the observed gains are not artifacts of open-ended evaluation metrics.

Notably, the qualitative behavior on GSM8K mirrors that on open-ended tasks: naive temperature scaling degrades accuracy without delivering meaningful diversity, while base–aligned collaboration enables diversity improvements with accuracy maintained. Complete results for both verifiable benchmarks are reported in Appendix §F.

**Multi-turn tasks.** To verify gains beyond single-turn settings, we evaluate on MT-Bench (Zheng et al., 2023). BACO-P-PUNC substantially outperforms the aligned baseline (Coverage: 0.681 vs. 0.320; Dominance: 72.8% vs. 27.2%) consistently across diversity and quality metrics; full results in Appendix F.

**Examination on the distribution of repetition.** We additionally evaluate BACO on Artificial Hivemind (Jiang et al., 2025), which measures intra-model repetition on open-ended chat queries. Following its setting, we compare BACO against Top-$p$ and Min-$p$ (Nguyen et al., 2025) on Llama-3.1-70B-Instruct.

With BACO's threshold selected to match Min-$p$'s quality (avg. reward 8.08; $\gamma$=0.2, -P-PUNC), its repetition distribution is flatter (Figure 5), placing more (15%) generations in the genuinely diverse regime (similarity < 0.6), vs. 8% for

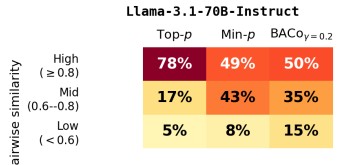

*Figure 5.* Pairwise intra-model repetition distribution under Artificial Hivemind. BACO's generation falls more in lower-similarity bins (bottom rows).

| Method | Lexical | | Semantic | | Overall | |
| --- | --- | --- | --- | --- | --- | --- |
| | *Cov.* | *Dom.* | *Cov.* | *Dom.* | *Cov.* | *Dom.* |
| Base | 0.142 | 9.8% | 0.142 | 13.2% | 0.142 | 11.5% |
| Aligned | 0.273 | **43.8%** | 0.128 | 19.9% | 0.200 | 31.8% |
| AACO | 0.022 | 4.5% | 0.006 | 7.0% | 0.014 | 5.7% |
| BACO | **0.495** | 42.0% | **0.452** | **59.8%** | **0.474** | **50.9%** |

*Table 4.* Comparison of base-aligned and aligned-aligned collaboration (denoted as "AACO") on NoveltyBench across all diversity–quality spaces. AACO yields little improvement on the diversity–quality trade-off compared with BACO, particularly in semantic diversity. The results demonstrate the necessity of involving a base and aligned model.

Min-$p$ and 5% for Top-$p$. This suggests BACO pushes diversity beyond the limit reachable by sampling within a single aligned model.

## 5.2. Router Performance Comparison

We analyze routers on NoveltyBench, which provides a representative yet computationally efficient setting for controlled comparison.

**Sanity-check routers (-RAND, -JUDGE).** At first glance, -RAND appears competitive on aggregate, but its gains concentrate almost entirely on lexical metrics (e.g., Dist-$n$) and collapse on semantic metrics such as Semantic Entropy (0% Dominance; Appendix §G). This is consistent with the observation that random or nonsensical text can inflate lexical diversity without producing semantically meaningful variation; unguided switching injects surface-level randomness rather than meaningful semantic diversity. By contrast, principled routers such as -P-PUNC achieve clear Dominance on both lexical and semantic spaces (Table 3). We further consider -JUDGE, a prompt-based router inspired by multi-agent systems (Talebirad & Nadiri, 2023). Despite extensive prompt engineering[9], it consistently underperforms simpler heuristic routers while incurring substantially higher computational cost.

**Simple routers (-P, -FC).** Single-strategy routers based on uncertainty (-P) or linguistic cues (-FC) trade weaker aggregate performance for clear gains in semantic diversity. Both outperform -RAND on metrics such as Cosine Dissimilarity and Semantic Entropy, demonstrating that informed routing is critical for meaningful diversity.

**Combining strategies yields the strongest performance.** Combining complementary strategies proves most effective.

---

[9]Including a step-by-step decision pipeline, curated heuristic rules, and few-shot examples with rationales (prompts are in Table A17).

In particular, -P-PUNC achieves the best overall controllability, with high Coverage and Dominance across both lexical and semantic spaces. Other combinations (-H-PUNC, -P-FC) capture additional regions of the Pareto frontier (17.5% and 17.6% Dom.), confirming that compositional routing provides complementary benefits.

Overall, BACO with most of the routers outperforms the baselines. We therefore emphasize that the contribution of BACO is the *framework* of token-level base-aligned collaboration, not any single router. And the framework is robust to router choice without heavy tuning.

## 5.3. Base-Aligned vs. Aligned-Aligned Collaboration

We compare base-aligned collaboration (BACO, `Llama-3-8B` family) with aligned-aligned collaboration (`Llama-3-8B` and `Llama-3.2-11B`) under identical router (-P-PUNC). On NoveltyBench, base-aligned collaboration significantly outperforms the aligned-aligned setup in both Coverage and Dominance. These results highlight the limited diversity achievable when collaborating between two aligned models and support our hypothesis that complementarity between less-aligned and more-aligned models are essential for jointly optimizing diversity and quality.

## 5.4. Long-Form Diversity Evaluation

To assess diversity beyond short-form outputs, we evaluate long-form creative writing using a discourse-level framework following Tian et al. (2024). Rather than relying on surface lexical or single-vector semantic metrics, this evaluation captures narrative variation through plot structure and affective dynamics (detailed setups are provided in Appendix H.1).

As shown in Figure 6, BACO achieves substantially higher turning-point and arousal diversity than the aligned baseline at comparable quality, demonstrating that base–aligned collaboration extends effectively to long-form generation. Example outputs are provided in Table A20.

## 5.5. Human Evaluation

To complement automatic metrics, we conduct a three-phase human evaluation assessing quality, group-level diversity, and creativity. Four expert annotators evaluate outputs on NoveltyBench and WildChat.

As shown in Table 5, despite near-identical automatic quality scores (aligned: 5.93; BACO: 5.85), human raters assign BACO substantially higher quality ratings (4.04 vs. 2.83 on NoveltyBench; 3.83 vs. 3.44 on WildChat), with strong inter-rater agreement (Pearson $r=0.816$, ICC$(2, k)=0.907$). Meanwhile, BACO achieves significant large diversity win rates across both format and content sub-dimensions (Fleiss'

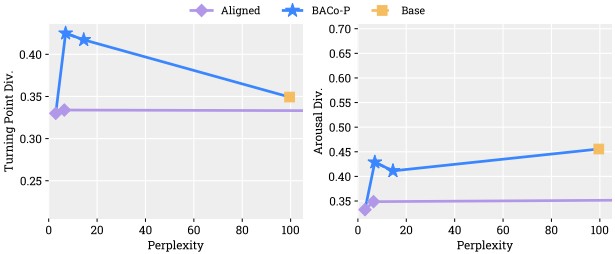

*Figure 6.* Comparison of BACO's and baselines' discourse-level diversity-quality trade-off curve on Narrative Discourse. BACO obtains a larger Coverage, achievable in the high-diversity, high-quality region (top-left). The results demonstrate it has richer discourse-level diversity without sacrificing quality largely, compared with baselines. The x-axis is quality (perplexity; lower is better), and the y-axis is discourse-level diversity, either turning-point diversity (left figure) or arousal diversity (right figure) (higher is better).

| | NoveltyBench | | WildChat | |
| --- | --- | --- | --- | --- |
| | Aligned | BACO | Aligned | BACO |
| *Quality* | 2.83 | **4.04** | 3.44 | **3.83** |
| *Diversity* *Overall* | 21.0% | **79.0%** | 36.1% | **63.9%** |
| *Format* | 25.4% | **74.6%** | 26.8% | **73.2%** |
| *Content* | 22.9% | **77.1%** | 41.6% | **58.4%** |
| *Creativity* | 20.4% | **79.6%** | 38.2% | **61.8%** |

*Table 5.* Human evaluation comparing BACO (best variant, -P-PUNC) with the aligned model baseline on NoveltyBench and WildChat. BACO is consistently preferred by human judges across all aspects, demonstrating that it produces outputs that are not only more human-perceived diverse but also higher in quality and creativity. Scores of quality are on a 1–5 Likert scale, and others indicate the pairwise win rate.

$\kappa$=0.268 overall), confirming that BACO produces human-perceivable diversity at both levels. Full protocols and annotation examples are provided in Appendix §J.

**From diversity-quality trade-off to creativity.** Building on Jaarsveld et al. (2012), creativity requires both *divergent thinking*, supported by group-level output diversity, and *convergent thinking*, underpinned by per-output quality. We preliminarily test whether an improved diversity-quality trade-off translates into higher human-perceived creativity using BACO. As shown in Table 5, BACO's outputs are judged most creative in 79.6% of NoveltyBench prompts and 61.8% of WildChat prompts (Fleiss' $\kappa$=0.485), suggesting that easing the diversity-quality trade-off yields outputs humans find genuinely more creative. We leave a larger-scale and deeper investigation of this connection to future work.

## 6. Analysis and Discussion

### 6.1. Contribution Distribution and Switching Frequency

For BACO with the best router (-P-PUNC), we see a consist pattern that base-model contribution and switching frequency are high at the start of generation and decrease over time across all three tasks (Figures A9-A11 in Appendix I). We hypothesize this reflects increasing predictive confidence as context grows, which naturally aligns with, e.g., in creative writing, early tokens allow more divergence (e.g., introducing characters) while later tokens demand coherence. However, the pattern may be less suitable for list-structured tasks that benefit from uniform exploration level over time. Dynamic, position-aware thresholds are a promising mitigation.

### 6.2. Failure Mode: Inherent Early Stopping

When the router is tuned more aggressively toward diversity, the system exhibits a higher tendency to terminate early. One model emits an end-of-sequence token prematurely (examples in Appendix I). This *inherent early stopping* is emergent rather than deliberate, unlike early-exit mechanisms in reasoning systems (Ding et al., 2025a). We take a two-sided view of this trend: it can risk truncating valid continuations, but also acts as a safeguard against incoherent repetition or off-topic continuation. Truncated outputs are easily detectable by length, allowing a simple restart strategy as mitigation with marginal cost overhead.

### 6.3. Future Work

We envision BACO as a preliminary step toward a "breadth thinking" mode for LLMs, complementing deep thinking (OpenAI, 2025; DeepSeek-AI et al., 2025), where a model explores a wide, validated space of perspectives rather than converging on a single output. For open-ended tasks, breadth thinking offers a way to help humans break out of their "information cocoons" (Piao et al., 2023), expand ideation space, and think beyond conventional boundaries. BACO is controllable, inference-time only, and deployable without additional post-training; it can also serve as a plug-in for broader systems, such as an ideation agent in multi-agent collaboration (Siddiqui et al., 2025; Cheng et al., 2025; Song et al., 2025). The controllability of BACO also opens opportunities for user-facing interfaces, such as a slider that lets users shift along the diversity-quality spectrum, offering a more principled way to control output diversity than temperature scaling, which degrades might quality.

## Acknowledgment

We thank the anonymous reviewers and area chair for their constructive comments, which helped improve this work. We also thank the members of the Communication and Intelligence (C&I) group at the University of Chicago for valuable discussions and proofreading. We thank Peter West (Stanford; University of British Columbia) for helpful discussions on the idea. We are grateful to the UChicago PhD Writing Program for their feedback on polishing the writing.

## Impact Statement

This work does not involve any sensitive personal data or personally identifiable information in the human study. All datasets used are publicly available research benchmarks released under appropriate licenses. Our experiments focus on inference time with existing LLMs without additional fine-tuning, and thus pose minimal risks of memorization or privacy leakage beyond those already associated with the underlying models. While our method aims to improve diversity in model outputs, we acknowledge the possibility that enhanced diversity might amplify harmful generations if safeguards are not in place. This is a shared risk for most diversity-promoting technologies. We therefore emphasize that BACo is complementary to safety alignment and should be deployed with standard content-filtering or moderation practices. Importantly, BACo is designed for open-ended generation, rather than high-stakes or safety-critical domains. All authors have read and adhered to the ICML Code of Ethics.

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

## Limitations

We see several limitations in our work.

(*i*) Collaboration between two models, by nature, incurs additional FLOPs overhead per token relative to single-model inference; the prototype is not yet perfectly engineered for speed, but we see speculative decoding, LoRA-based aligned models, and KV-cache sharing are mitigations (Appendix I).

(*ii*) Beyond the open-ended generation tasks studied here, other tasks such as agent and code generation can also benefit from diversity, and can extend to much longer horizons; we leave these extensions for future work.

(*iii*) Our focus is inference-time improvement, so directions such as learned routers and leveraging diversity within RL rollouts, while relevant, are out of scope for this paper.

(*iv*) We mainly study the canonical base and aligned checkpoints; intermediate alignment checkpoints, where available, likely offer additional Pareto gains and are a promising direction (Appendix I).

## A. Related Work

**Diversity Degradation in Alignment.** While alignment techniques like RLHF enhance model performance in instruction following and reasoning, they systematically curtail output diversity. This trade-off is widely recognized, with a growing body of evidence demonstrating that aligned models are less diverse than their base counterparts. For example, studies have questioned their artistic authenticity (Chakrabarty et al., 2024), and benchmarks like NoveltyBench reveal their diminished capacity for humanlike diversity and creativity (Zhang et al., 2025; Tian et al., 2024; Lu et al., 2025; West & Potts, 2025). The underlying mechanism for this degradation is "probability concentration," as the alignment process "sharpens" the model's output distribution, thereby steering it into low-entropy generation paths (Yang & Holtzman, 2025). The diversity degradation impacts the downstream applications. It manifests as a loss of linguistic idiosyncrasies (Chakrabarty et al., 2025), increased format homogeneity (Zhang et al., 2024), and diminished creativity (West & Potts, 2025) in generated text. Beyond linguistic characteristics, alignment-induced constraints lead models to converge on a restricted repertoire of strategies, thereby diminishing diversity in reasoning (Chen et al., 2024; Ding et al., 2025b), data synthesis (Kim et al., 2024; Yang et al., 2025), deep research (Xiao, 2025), social simulation (Wang et al., 2025a), and gaming (West & Potts, 2025). More critically, the lack of output diversity has been shown to further reduce outcome diversity (Padmakumar & He, 2023) and creativity (Meincke et al., 2025; Ashkinaze et al., 2025) in human interactions with these models. These studies demonstrate that diminished diversity in model outputs adversely affects how humans ideate, create, and engage. Our work addresses this challenge directly, proposing an inference-time collaborative decoding framework that optimizes the diversity-quality trade-off by combining the strengths of both base and aligned models.

**Diversity-Promoting Methods.** Approaches to enhance the diversity of aligned LLMs fall into two main categories: training-time and inference-time methods. **Training-time methods** typically modify the learning objective to encourage varied outputs. A prominent line of work adapts Direct Preference Optimization (DPO; Rafailov et al. (2023)) by incorporating diversity-aware mechanisms, such as f-divergence penalties (Wang et al., 2024), set-level diversity rewards (Lanchantin et al., 2025), or re-weighted loss objectives (Chung et al., 2025; Ismayilzada et al., 2025). Other approaches leverage different architectures, like generative flow networks, to the same end (Kwon et al., 2024). While these methods can instill diversity directly into the model, they require substantial computational resources for retraining and offer little flexibility for user-specific diversity needs at inference. **Inference-time methods** offer a more lightweight and adaptable alternative. These include modifications to decoding algorithms like diverse beam search (Vijayakumar et al., 2016) and various prompt engineering strategies, such as paraphrasing (Meyerson et al., 2024; Wang et al., 2025b; Zhang et al., 2025; Wang et al., 2025a; Wong et al., 2024). Existing inference-time methods for improving diversity typically incur high computational costs via multiple decoding passes or long-horizon planning. Or they significantly degrade generation quality (Peeperkorn et al., 2024). Achieving a stable diversity-quality trade-off with these techniques remains a challenge. Our proposed method BACo, is an inference-time framework designed to offer more explicit and reliable control over this trade-off. We therefore focus our comparison on baselines from this category.

**Multi-Model Collaborative Generation.** Prior work has explored collaborative frameworks where multiple language models work in concert to improve generation quality and efficiency (e.g., computation cost and latency). These approaches can be grouped by their collaboration mechanism. One line of work focuses on **weight-level collaobration**. This includes

merging reward or policy models, or using Mixture-of-Experts (MoE) (Shazeer et al., 2017) architectures to create a single, more capable system better aligned with diverse human preferences (Rame et al., 2023; Zheng et al., 2025; Shi et al., 2025). Another mechanism is **token-level collaboration**, where multiple models collaborate during decoding by exchanging next-token probability distributions or candidate token choices at each step, to improve attributes like coherence and factuality, or reduce latency (Leviathan et al., 2023a; Li et al., 2023; Zheng et al., 2024; Fei et al., 2025). More recently, **multi-agent systems** have emerged, in which models debate or discuss to leverage their complementary strengths for complex, creative tasks (Lu et al., 2024; Venkatraman et al., 2025; Huot et al., 2025). Our BACo, advances token-level collaboration. While existing methods in this area primarily target quality or efficiency, we focus on navigating the diversity-quality trade-off.

## B. BACo Framework Implementation and Router Details

**Models.**    We apply each model's default chat template during collaboration (for base models, we apply a plain shifting template), served on vLLM local host. We disable every model's tool call and thinking for fair comparison.

**Tokenization Alignment.**    Tokenization sometimes mismatches between the two models (even base and aligned pairs), particularly around punctuation, special tokens, or rare words, can lead to incoherent sub-word boundaries. To address this, we enforce that tokens representing a single semantic unit (e.g., a word or format element) must all be produced by the same model. This avoids artifacts such as broken punctuation or malformed words.

**Algorithm Overview.**    Algorithm 1 formalizes the full BACo decoding procedure, including the additional rules described above (aligned first token, word-boundary switching, aligned-controlled EOS) and reconciles Eq. equation 1 with the actual sampling step. We express the router as an ordered tuple of routing strategies $\mathcal{R} = (\mathcal{R}_1, \ldots, \mathcal{R}_K)$, where each $\mathcal{R}_k$ may fire in either direction (base or aligned) on its trigger condition, or abstain otherwise (move to next strategy). At a routing step, the strategies are evaluated in order and the first $\mathcal{R}_k$ that fires determines $m$. By construction, the last strategy in the tuple always fires, so the cascade always terminates with a definite decision. Single-strategy routers correspond to $K = 1$; combination routers correspond to $K \geq 2$ with complementary signals in practice.

---

**Algorithm 1** BACo Decoding

---

**Require:**  prompt $x$; base model $P_{\text{base}}$; aligned model $P_{\text{aligned}}$; router $\mathcal{R} = (\mathcal{R}_1, \ldots, \mathcal{R}_K)$ with threshold $\gamma$; max length $T$
1:  $y_1 \sim P_{\text{aligned}}(\cdot \mid x)$                                          ▷ sample first token from aligned
2:  $m \leftarrow$ aligned
3:  **for** $t = 2, \ldots, T$ **do**
4:      $c_t \leftarrow [x, y_{<t}]$
5:      **if** $y_{t-1}$ ends a word **then**
6:          $m \leftarrow \mathcal{R}(c_t, P_{\text{base}}, P_{\text{aligned}}, \gamma)$          ▷ go through each strategies within router $\mathcal{R}$ and returns base or aligned
7:      **end if**
8:      $y_t \sim P_m(\cdot \mid c_t)$
9:      **if** $y_t = \langle\text{eos}\rangle$ **and** $\arg\max P_{\text{aligned}}(\cdot \mid c_t) \neq \langle\text{eos}\rangle$ **then**
10:          resample $y_t \sim P_m(\cdot \mid c_t)$ excluding $\langle\text{eos}\rangle$                  ▷ aligned-controlled termination
11:      **end if**
12:      **if** $y_t = \langle\text{eos}\rangle$ **then**
13:          **break**
14:      **end if**
15:  **end for**
16:  **return** $y_1, \ldots, y_t$

---

**Framework Cost.**    BACo requires two forward passes per decoding step at the worst case, incurring $\sim 2\times$ FLOPs overhead per token regardless of sample count. Crucially, this overhead is independent of the desired group size $n$, in contrast to baselines such as in-context resampling and paraphrase prompting (which require $n$ sequential passes) or diverse beam search (which expands $\geq n$ beams). Moreover, stems from *superficial alignment* (Zhou et al., 2023a; Lin et al., 2023), interventions between the largely-agreeing base and aligned models can be sparse (Fei et al., 2025). This sparsity enables practical optimizations such as caching multi-token chunks from one model to minimize switching costs and overlapping communication with computation to hide context-switching latency. Further engineering-level optimizations

are complementary and applicable when deploying at scale, including speculative decoding (Leviathan et al., 2023a), LoRA-based aligned models (Hu et al., 2021), and KV cache sharing (Liu et al., 2025).

**Wall-Clock Runtime Comparison.**    We empirically measure runtime under identical hardware (single A100 80 GB) for inference-time methods, using the HuggingFace library across the board for fair comparison. Results are in Table A6. BACO's current prototype is slower than single-model baselines but is comparable to or faster than several diversity-promoting baselines (e.g., Diverse Beam Search, Logits Ensemble) while substantially outperforming them on the diversity-quality trade-off. The implementation is not yet engineered for speed; the optimizations listed above can further reduce wall-clock cost.

| Method | Time/Sample (s) | Tok/s | Peak Memory |
|---|---|---|---|
| Aligned | $9.8 \pm 3.9$ | 40.0 | 15.1 GiB |
| Response Ensemble | $10.9 \pm 4.0$ | 34.8 | 15.4 GiB |
| Diverse Beam Search | $17.0 \pm 5.9$ | 17.0 | 15.8 GiB |
| Logits Ensemble | $25.5 \pm 0.5$ | 20.1 | 30.1 GiB |
| In-Context Resampling | $90.5 \pm 35.3$ | 4.1 | 15.5 GiB |
| Paraphrase Prompting | $117.9 \pm 41.1$ | 3.3 | 15.1 GiB |
| Back Translation | $120.5 \pm 65.6$ | 3.8 | 15.1 GiB |
| Mix-of-Agents | $224.9 \pm 123.2$ | 3.8 | 15.6 GiB |
| Multi-Agent Debate | $239.0 \pm 81.5$ | 3.5 | 15.6 GiB |
| BACO | $22.9 \pm 4.5$ | 19.2 | 29.2 GiB |

*Table A6.* Wall-clock runtime comparison on a single A100 80 GB. Tok/s is computed over the final output. BACO is comparable to or faster than several inference-time diversity baselines while substantially outperforming them on diversity–quality (Table 2).

## B.1. Additional Rules in Router

We follow Fei et al. (2025) in always using the aligned model to generate the first token. Early decoding steps have an outsized influence on generation and typically show greater disagreement between models. Starting from the aligned model improves trajectory quality and reduces the chance of degenerate completions.

Incorporating low-probability tokens from the base model introduces new challenges in sequential generation. When switch between models, the receiving model may struggle to continue from an unfamiliar context. In particular, the aligned model may terminate the output prematurely, while the base model may fall into degenerate behaviors such as repetition or verbose listing. To mitigate this, we constrain output termination by only accepting the end-of-sentence token when it is the top-1 prediction of the aligned model.

## B.2. All Routing Strategies and Notations

The following are all strategies and their corresponding notations that are mentioned in this paper:

- -RAND: Route to the base model by random chance $\gamma$.

- -P: Route to the base model when base model's top-1 token probability $\max_{y_t} P_{\text{base}}(y_t \mid x, y_{<t}) < \gamma$ $_{\gamma \in [0,1]}$, otherwise to the aligned model.

- -P-A: Route to the base model when aligned model's top-1 token probability $\max_{y_t} P_{\text{aligned}}(y_t \mid x, y_{<t}) < \gamma$ $_{\gamma \in [0,1]}$, otherwise to the aligned model.

- -H: Route to the base model when the entropy of the base model's next token prediction distribution $H_{\text{base}}(y_t \mid x, y_{<t}) > \gamma$ $_{\gamma \in [0,+\infty)}$, otherwise to the aligned model.

- -H-A: Route to the base model when the entropy of the aligned model's next token prediction distribution $H_{\text{aligned}}(y_t \mid x, y_{<t}) > \gamma$ $_{\gamma \in [0,+\infty)}$, otherwise to the aligned model.

- -PR: Route to the base model when the ratio between the base model's top 1 token probability and the aligned model's, i.e., $\frac{\max_{y_t} P_{\text{base}}(y_t|x,y_{<t})}{\max_{y_t} P_{\text{aligned}}(y_t|x,y_{<t})} < \gamma$ $_{\gamma \in (0,1]}$, otherwise to the aligned model.

- -HR: Route to the base model when the ratio between the base model's entropy and the aligned model's, i.e., $\frac{H_{\text{base}}(c_i|q,c_{<i})}{H_{\text{aligned}}(c_i|q,c_{<i})} > \gamma$ $_{\gamma \in [1,+\infty)}$, otherwise to the aligned model.

- -FC: Route to the base model when both the aligned model and base model sample the next token is a content word[10], otherwise to the aligned model.

- -PUNC: Route to the base model when the base model's and the aligned model's next token is not punctuation or formative tokens (e.g., '$\backslash n$'), otherwise, to the aligned model.

- -JUDGE: Route to the base model when an external judge LLM (another aligned model) determines that both of the following conditions are satisfied: *1)* the next token continuation has space to diverge; *2)* the sampled base model continuation is acceptable (i.e., reasonable and meaningful). Otherwise, to the aligned model.

Comparatively, we observe that the aligned model's logits-based metrics are less distinctive, which aligns with the literature on entropy decrease, hence making routing strategies such as -P-A and -H-A less effective compared with the same metric under the base model's logits.

-FC and -PUNC fail under the same motivation of using content-based linguistic features as a routing strategy, where -PUNC is more lightweight than -FC. From our empirical observation, the two have on-tier performance. However, the introduction of part-of-speech parsing for -FC takes additional computational cost. Given the cost limitation, some experiments and analyses take -PUNC as representative.

We prompt -JUDGE with curated heuristic rules and few-shot examples with rationals. Detailed prompt designs are at Table A17. As a more costly strategy, it serves as an extended comparison.

Following on, we have multi-condition routers, which are some possible combinations of the above single-condition routers:

- -P-FC: First apply the -FC rule and then the -P. Route to the base model when any one of the following conditions is met: *1)* base model sampled next token is a function word; *2)* both models' sampled next token are a content word; *3)* base model's top-1 token probability $\max_{y_t} P_{\text{base}}(y_t \mid x, y_{<t}) < \gamma$ $_{\gamma \in [0,1]}$. Otherwise, to the aligned model.

- -P-PUNC: First apply the -PUNC rule and then the -P. Route to the base model when any one of the following conditions is met: *1)* base model sampled next token is not a punctuation or formatting tokens; *2)* base model's top-1 token probability $\max_{y_t} P_{\text{base}}(y_t \mid x, y_{<t}) < \gamma$ $_{\gamma \in [0,1]}$. Otherwise, to the aligned model.

- -H-FC: First apply the -FC rule and then the -H. Route to the base model when any one of the following conditions is met: *1)* base model sampled next token is a function word; *2)* both models' sampled next token are a content word; *3)* base model's entropy of next token prediction distribution $H_{\text{base}}(y_t \mid x, y_{<t}) > \gamma$ $_{\gamma \in [0,+\infty)}$. Otherwise, to the aligned model.

- -H-PUNC: First apply the -PUNC rule and then the -P. Route to the base model when any one of the following conditions is met: *1)* base model sampled next token is not a punctuation or formatting tokens; *2)* base model's entropy of next token prediction distribution $H_{\text{base}}(y_t \mid x, y_{<t}) > \gamma$ $_{\gamma \in [0,+\infty)}$. Otherwise, to the aligned model.

## C. Dataset Details

**NoveltyBench** is a human-curated benchmark designed to evaluate the ability of LLMs to produce multiple distinct yet high-quality outputs. The instructions are constructed such that multiple valid answers exist, spanning four categories: randomness (e.g., "the result of a die roll"), underspecified factual knowledge (e.g., "tell me a capital city in Africa"), creative writing (e.g., "short poem or story"), and subjective queries (e.g., "recommendation or opinion"). While effective for fine-grained diversity evaluation, NoveltyBench prompts are intentionally simple and often yield short outputs with limited opportunities for variation. We therefore complement it with more complex datasets.

---

[10]Long words could be composed by multiple tokens. If so, we will route to the same model multiple steps until the word is finished.

**WildChat**   is a large-scale dataset of real human–LLM conversations. Following Zhang et al. (2025), we select a subset of prompts without fixed ground-truth answers to emphasize open-endedness, enabling evaluation under more realistic and challenging settings.

**Narrative-Discourse**   is a dataset for long-form creative writing, where models extend fictional film synopses in English. The dataset provides structured annotations of discourse-level elements such as turning points, story arcs, and arousal (Tian et al., 2024), enabling evaluation of long-form structural diversity.

# D. Automation Evaluation Details

## D.1. Diversity Metrics

We are mainly interested in diversity across a group of outputs. For each prompt $x$, we sample $n = 10$ outputs: $\{y_0, \ldots, y_{n-1}\}$. We evaluate the diversity of $\{y_i\}$ using a broad set of automated metrics. Below are full derivations and definitions, grouped by category.

### D.1.1. LEXICAL METRICS

**Distinct-$n$.**   Ratio of unique $n$-grams to total $n$-grams. Let $\mathcal{G}_n = \bigcup_{i=0}^{n-1} \text{ngrams}(y_i, n)$, tokenized by NLTK word-tokenize.

$$\text{Distinct-}n = \frac{|\text{set}(\mathcal{G}_n)|}{|\mathcal{G}_n| + \varepsilon} \in [0, 1]$$

Higher values indicate higher lexical diversity.

**Expectation-Adjusted Distinct (EAD-$n$).**   A length- and vocabulary-normalized variant of Distinct-$n$, mitigating bias from long outputs. Define $V$ as the $n$ power of the vocabulary size $V$ of the aligned model's tokenizer and the union of all $n$-grams similarly by the aligned model's tokenizer.

$$V = \text{vocabulary size}^n$$

$$\text{EAD-}n = \frac{|\text{set}(\mathcal{G}_n)|}{V \cdot \left(1 - \left(\frac{V-1}{V}\right)^{|\mathcal{G}_n|}\right) + \varepsilon} \in [0, 1]$$

Higher values indicate higher lexical diversity.

**Self-BLEU.**   Average pairwise BLEU (Papineni et al., 2002). For each output $y_i$, use the other outputs $\{y_j\}_{j \neq i}$ as references:

$$\text{Self-BLEU} = \frac{1}{n} \sum_{i=0}^{n-1} \text{BLEU}(y_i, \{y_j\}_{j \neq i}) \in [0, 1]$$

Lower values indicate higher lexical diversity.

**Self-ROUGE-L.**   Average pairwise ROUGE-L scores (Lin, 2004).

$$\text{Self-ROUGE-L} = \frac{1}{n} \sum_{i=0}^{n-1} \text{ROUGE-L}(y_i, \{y_j\}_{j \neq i}) \in [0, 1]$$

Lower values indicate higher lexical diversity.

### D.1.2. SEMANTIC METRICS

**Embedding Cosine Dissimilarity.**   Embed each $y_i$ using a sentence embedding model (e.g., SBERT or Qwen3), obtaining $\mathbf{e}_i$. Compute pairwise cosine distances:

$$d_{ij} = 1 - \cos(\mathbf{e}_i, \mathbf{e}_j)$$

$$\text{Embedding Diversity} = \frac{2}{n(n-1)} \sum_{i<j} d_{ij} \in [0, 1]$$

Higher values indicate higher semantic diversity.

**Vendi Score.** The exponential entropy of eigenvalues of the similarity matrix based on n-gram Jaccard overlap, capturing the effective number of independent modes. First, construct a similarity matrix $K \in \mathbb{R}^{n \times n}$ via either n-gram Jaccard overlap or pairwise embedding similarity, which is positive semi-definite. Let $\lambda_1, \ldots, \lambda_n$ be the eigenvalues of $\frac{K}{n}$. Then:

$$\text{Vendi Score} = \exp\left(-\sum_{i=1}^{n} \lambda_i \log \lambda_i\right) \in [1, n]$$

This is the exponential of the Shannon entropy of the normalized similarity matrix, interpretable as the effective number of distinct modes. We construct similarity matrix based on SimCSE embeddings (Gao et al., 2021). Higher values indicate higher semantic diversity.

**NLI Diversity.** Average contradiction probability across output pairs, computed using a RoBERTa NLI model.[11] For each pair $(y_i, y_j)$, apply an NLI model (RoBERTa-based) to compute the entailment probability:

$$\text{NLI Diversity} = \frac{2}{n(n-1)} \sum_{i<j} P_{\text{entailment}}(y_i, y_j) \in [0, 1]$$

Lower values (less entailment) indicate higher diversity.

**Distinct Score (NoveltyBench).** The number of unique functional equivalence classes predicted by a DeBERTa classifier trained on human annotation; The DeBERTa classifier is trained to predict whether two outputs are functionally equivalent. Cluster the outputs $\{y_i\}$ equivalence classes. The metric is:

$$\text{Distinctivity Score} = \#\{\text{unique equivalence classes among } \{y_i\}\} \in [0, n-1]$$

Larger values indicate higher diversity.

**Semantic Entropy.** Rao's quadratic entropy over clusters of semantically equivalent outputs grouped via entailment and aggregated via log-likelihood. It works by first clustering outputs $\{y_i\}$ into semantic groups $\{C_1, \ldots, C_k\}$ using entailment-based NLI. Then compute cluster-level probabilities using likelihoods:

$$\log p(C_k) = \log\left(\sum_{y_i \in C_k} \exp(\log p(y_i))\right)$$

Finally:

$$\text{Semantic Entropy} = -\sum_k p(C_k) \log p(C_k) \in [0, \log n]$$

Larger values indicate higher diversity.

### D.2. Aggregate Metrics over Quality–Diversity Spaces

Let a *space* be defined by a pair of metrics $(m_x, m_y)$, where $m_x$ measures quality (higher is better) and $m_y$ measures diversity (either higher or lower is better, depending on the metric). Varying a method's control parameter (e.g., decoding temperature, routing threshold) traces a set of points $\{(x_t, y_t)\}$ in this space.

**Feasible Region and Normalization.** To make values comparable across metrics, we normalize each space to the unit square $[0, 1]^2$. Let $\mathcal{F} = [x_{\min}, x_{\max}] \times [y_{\min}, y_{\max}]$ denote the feasible region, anchored using two reference operating points at temperature 1.0: the *base* model and the *aligned* model.[12] Observed points are normalized via:

$$\hat{x} = \frac{x - x_{\min}}{x_{\max} - x_{\min}}, \qquad \hat{y} = \begin{cases} \dfrac{y - y_{\min}}{y_{\max} - y_{\min}}, & \text{if higher is better,} \\ 1 - \dfrac{y - y_{\min}}{y_{\max} - y_{\min}}, & \text{if lower is better.} \end{cases}$$

---

[11] https://huggingface.co/sentence-transformers/nli-roberta-base-v2

[12] Concretely, $x_{\min}$ is set to the base model's quality at $T = 1.0$; $x_{\max}$ to the aligned model's quality at $T = 1.0$. For the diversity axis, if higher is better we set $y_{\min}$ to the aligned model's diversity at $T = 1.0$ and $y_{\max}$ to the theoretical maximum (e.g., $\log N$ for Semantic Entropy with $N$ samples). If lower is better, we set $y_{\min}$ to the theoretical lower bound and $y_{\max}$ to the aligned model's diversity at $T = 1.0$.

Points outside $\mathcal{F}$ are discarded for aggregation, as the outputs of the represented setting might have limited usage. It has no strength, in terms of the metrics, compared with the two single-model baselines.

**Coverage (Cov.)** measures how effectively a method traverses the diversity-quality trade-off as its control parameters vary (e.g., decoding temperature of single-model baselines and threshold for BACo routers). The indicator is simplified from Hypervolume (HV) (Zheng et al., 2017) in multiobjective optimization problems. Concretely, we normalize each space into a unit square (anchored by the default baseline: base and aligned models at temperature 1.0), and compute the area under the curve (AUC) traced by the method's normalized points. Higher Coverage values indicate greater controllability, general good performance across different trade-off balances, and robustness across parameters. However, Coverage does not capture whether a method is *ever* optimal across different trade-off balances.

For method $k$, we consider the piecewise-linear curve obtained from its normalized points $\{(\hat{x}_t, \hat{y}_t)\}$ (ordered by $\hat{x}$), augmented with boundary points to close the curve inside $[0,1]^2$. We define:

$$\text{Coverage}_k(m_x, m_y) = \int_0^1 \hat{y}_k(\hat{x}) \, d\hat{x},$$

computed using the trapezoidal rule. Because the domain is fixed to $[0,1]$, Coverage $\in [0,1]$. Higher values indicate that the method maintains strong quality and diversity as its control parameters vary.

**Dominance (Dom.)** complements Coverage by capturing whether a method ever achieves optimality relative to others. We utilize the C-metric (Zitzler, 1999) to evaluate Dominance of pairwise comparison, which captures the portion of the frontier that one method dominates over the other one. In our problem, the portion is in terms of intervals along the diversity (denoted as -D) or quality (denoted as -Q) axes. Dom takes the harmonic mean of Dom-D and Dom-Q. For global comparison across all methods, we compute the global Pareto frontier across all methods. We apply the C-metric between each method and the global frontier, equivalent to the portion of the frontier attributed to the method.

We compute the global Pareto frontier $\mathcal{P}$ over the union of all methods' normalized points in a space. For each Pareto point, we assign an *interval of responsibility* along the diversity or quality axis by splitting at midpoints between adjacent frontier points. Summing these interval lengths for Pareto points contributed by method $k$ yields its coverage along that axis, normalized by the total frontier span:

$$\text{Dom-D}_k(m_x, m_y), \quad \text{Dom-Q}_k(m_x, m_y) \ \in [0,1].$$

We report a single Dominance score as their harmonic mean:

$$\text{Dom}_k(m_x, m_y) = \frac{2 \cdot \text{Dom-D}_k \cdot \text{Dom-Q}_k}{\text{Dom-D}_k + \text{Dom-Q}_k}$$

**Holistic aggregation.**   Since quality and diversity admit multiple measurements, we average over all spaces $\mathcal{S}$ to obtain metric-agnostic summaries:

$$\overline{\text{Cov}}_k = \frac{1}{|\mathcal{S}|} \sum_{(m_x, m_y) \in \mathcal{S}} \text{Cov}_k(m_x, m_y), \qquad \overline{\text{Dom}}_k = \frac{1}{|\mathcal{S}|} \sum_{(m_x, m_y) \in \mathcal{S}} \text{Dom}_k(m_x, m_y).$$

## E. Experiment Setup

**Inference Setup.**   Our study focuses on group-level diversity. For each prompt, we generate a group of $n = 10$ outputs. Unless otherwise specified, sampling uses temperature 1.0 and nucleus sampling (top-$p$) with $p = 0.9$ and no top-$k$ truncation, following Zhang et al. (2025). For Diverse Beam Search (Vijayakumar et al., 2016), we use beams=$2n$=20, beam_groups=$n$=10, and diversity_penalty=1.0, following the original paper. Other exceptions apply only to baselines that inherently require alternative decoding strategies.

## F. Validation on Verifiable Benchmarks

This section reports detailed results on two verifiable benchmarks used to validate that the improvements of BACo are not artifacts of open-ended evaluation metrics: verifiable instruction following (IFEval) and mathematical reasoning (GSM8K).

**Verifiable instruction following (IFEval).** IFEval provides instruction-following tasks with automatically verifiable constraints (Zhou et al., 2023b). We evaluate instruction-following accuracy as a quality metric alongside diversity metrics. As shown in Figure A7, at matched quality levels, BACo achieves consistently higher diversity than the aligned baseline.

**Mathematical reasoning (GSM8K).** For mathematical reasoning, we evaluate on GSM8K (Cobbe et al., 2021) and measure Acc and Pass@10 as the accuracy metrics. Diversity is computed over sampled solution outputs. As shown in Figure A8, BACo maintains high accuracy (e.g., 90% Pass@10) while achieving substantially higher diversity across a wide range of operating points.

Overall, the qualitative trends on both benchmarks mirror those observed on open-ended generation tasks, confirming that the gains of BACo generalize beyond open-ended evaluation metrics.

## G. Detailed Results on Open-ended Tasks

### G.1. Instruction Following on NoveltyBench

Table A7 shows the result on NoveltyBench of comparing BACo on the best router compared with baselines.

| Method | Lexical | | Semantic | | Overall | |
|---|---|---|---|---|---|---|
| | *Cov.* | *Dom.* | *Cov.* | *Dom.* | *Cov.* | *Dom.* |
| Base | 0.142 | 9.8% | 0.142 | 13.1% | 0.142 | 11.4% |
| Aligned | 0.273 | **40.1%** | 0.128 | 17.2% | 0.200 | 28.6% |
| Nudging | 0.192 | 6.8% | 0.161 | 7.6% | 0.176 | 7.2% |
| Decoding | - | 0.8% | - | 1.0% | - | 0.9% |
| Prompting best | - | 8.0% | - | 6.5% | - | 7.3% |
| Ensemble best | - | 3.4% | - | 5.8% | - | 4.6% |
| BACo best | **0.495** | 31.0% | **0.452** | **48.8%** | **0.474** | **39.9%** |

*Table A7.* Comparison results on NoveltyBench. For space-saving, we present the best method in each category.

**Results.** BACo outperforms all baselines on all metrics except lexical Dominance. Compared with all baselines, BACo improves Coverage by **0.274** overall (0.222 lexical, 0.291 semantic). It dominates **39.9%** (the most) of the diversity-quality frontier overall (31.0% lexical, 48.8% semantic).

Beyond the LLaMA-3 base–aligned pair, we also validate that BACo consistently outperforms baselines on another model family, Olmo2. The results are reported in Table A11.

### G.2. Dialogue on WildChat

WildChat involves naturally complex and nuanced prompts, leading to much longer outputs on average compared with NoveltyBench. Table A8 summarizes the results.

| Method | Lexical | | Semantic | | Overall | |
|---|---|---|---|---|---|---|
| | *Cov.* | *Dom.* | *Cov.* | *Dom.* | *Cov.* | *Dom.* |
| Base | 0.000 | 1.9% | 0.000 | 6.8% | 0.000 | 4.38% |
| Aligned | 0.253 | **59.2%** | 0.077 | 29.1% | 0.165 | **44.1%** |
| Nudging | 0.430 | 11.4% | 0.387 | 15.6% | 0.408 | 13.5% |
| BACo best | **0.473** | 27.4% | **0.454** | **48.5%** | **0.463** | 38.0% |

*Table A8.* Comparison results on WildChat. For space saving, we present the best router, -P-PUNC, as BACo's representative.

**Results.** The superiority of BACo persists on WildChat. Compared with the aligned model baseline, BACo-P-PUNC

| Method | Lexical | | Semantic | | Overall | |
|---|---|---|---|---|---|---|
| | *Cov.* | *Dom.* | *Cov.* | *Dom.* | *Cov.* | *Dom.* |
| Base | 0.151 | 26.3% | 0.153 | 28.1% | 0.152 | 27.2% |
| Aligned | 0.282 | **47.7%** | 0.106 | **41.2%** | 0.194 | **44.4%** |
| Nudging | 0.205 | 9.7% | 0.194 | 6.5% | 0.199 | 8.1% |
| BACo best | **0.367** | 16.3% | **0.174** | 24.2% | **0.271** | 20.3% |

*Table A9.* Comparison results on Narrative-Discourse. For space saving, we present the best router, -P-PUNC, as BACo's representative.

| Method | Lexical | | Semantic | | Overall | |
|---|---|---|---|---|---|---|
| | *Cov.* | *Dom.* | *Cov.* | *Dom.* | *Cov.* | *Dom.* |
| Base | 0.142 | 7.6% | 0.142 | 10.9% | 0.142 | 9.2% |
| Aligned | 0.273 | 36.5% | 0.128 | 15.8% | 0.200 | 26.1% |
| In-context Prompt | - | 0.0% | - | 2.2% | - | 1.1% |
| Paraphrase Prompt | - | 8.0% | - | 5.6% | - | 6.8% |
| Diverse BS Decoding | - | 0.8% | - | 1.0% | - | 0.9% |
| Response Ensemble | - | 3.4% | - | 3.7% | - | 3.6% |
| Logits Ensemble | - | 0.0% | - | 0.0% | - | 0.0% |
| Nudging | 0.192 | 4.9% | 0.161 | 4.0% | 0.176 | 4.5% |
| BACo All | 0.495 | 39.0% | 0.452 | 56.8% | 0.474 | 47.9% |
| BACo-JUDGE | 0.302 | 0.5% | 0.254 | 0.1% | 0.278 | 0.3% |
| BACo-RAND | 0.493 | 13.1% | 0.409 | 5.9% | 0.451 | 9.5% |
| BACo-FC | 0.419 | 2.4% | 0.382 | 4.0% | 0.401 | 3.2% |
| BACo-P | 0.433 | 2.6% | 0.397 | 7.7% | 0.415 | 5.2% |
| BACo-P-PUNC | 0.495 | 11.4% | 0.452 | 17.2% | 0.474 | 14.3% |
| BACo-H-PUNC | 0.466 | 5.9% | 0.427 | 11.0% | 0.446 | 8.4% |
| BACo-P-FC | 0.435 | 3.1% | 0.406 | 10.9% | 0.421 | 7.0% |

*Table A10.* Comparison of all methods (baselines and BACo routers) on NoveltyBench. BACo All reports the best *Cov.* across all routers and the *Dom.* sum over all routers. The lower half of the table provides the performance of individual routers. Note that routers distribute *Dom.* values given to the metric definition; therefore, *Dom.* values in the top and bottom halves of the table are not directly comparable.

improves Coverage by 29.8% and dominates 30.8% of the frontier. Moreover, BACo demonstrates a particularly strong advantage in semantic diversity, where it dominates 48.5% of the frontier. These findings confirm that base–aligned collaboration scales effectively from short-form prompts (NoveltyBench) to longer, more conversational dialogue.

## G.3. Creative Writing on Narrative-Discourse

We further evaluate BACo on Narrative-Discourse to test its ability to generate structure-diverse and long-term coherent narratives. This dataset emphasizes sustained creativity and narrative arc, placing distinct demands beyond instruction following and dialogue.

**Results.** As shown in Table A9, BACo again outperforms all baselines. It achieves 13.5% higher Coverage and dominates 20.3% of the overall diversity–quality frontier.

## G.4. Extensional Tasks Results

We compare BACo-P-PUNC (i.e., our best router) with the aligned model baseline by adjusting temperature. Figure A7 presents the full results of the verifiable instruction following task on the IFEval dataset, and Figure A8 presents the full results of the mathematical reasoning task on the GSM8K dataset.

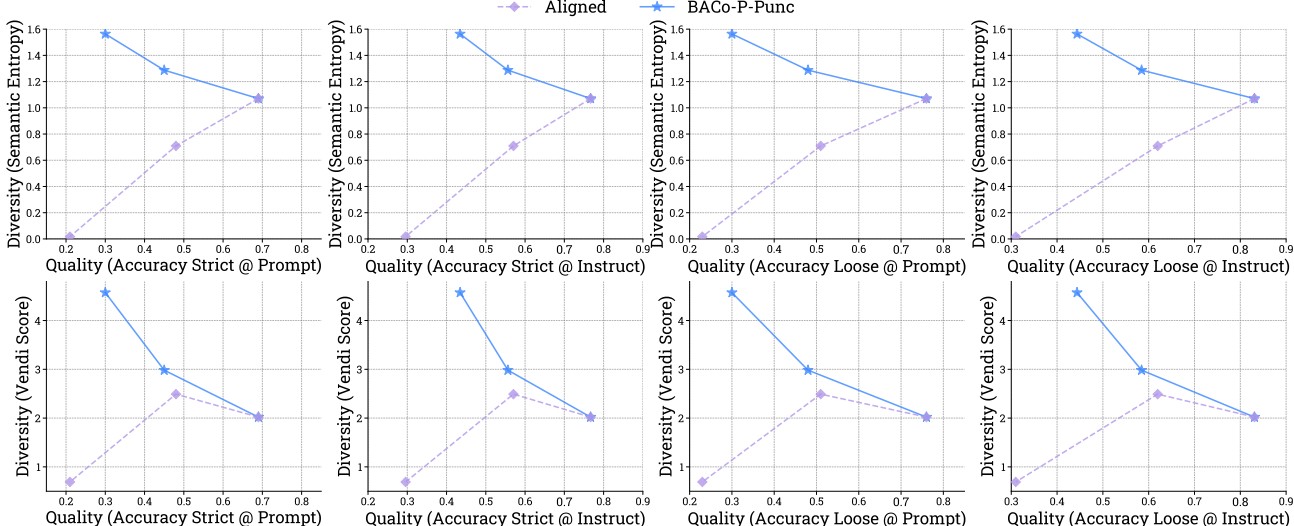

*Figure A7.* Diversity-quality trade-off comparison between BACO and the aligned model baseline on IFEval. BACO demonstrates a more effective optimization of the trade-off. The table presents all the trade-off spaces tested.

| Method | Overall *Cov.* | Overall *Dom.* |
|---|---|---|
| Base | 0.098 | 11.7% |
| Aligned | 0.209 | 30.5% |
| Nudging | **0.281** | 5.5% |
| Others | - | 8.3% |
| BACO-P-PUNC | 0.236 | **44.0%** |

*Table A11.* Results of BACO on `Olmo2-7B` and `Olmo2-7B-Instruct` on NoveltyBench.

# H. Narrative Discourse Evaluation Details

## H.1. Creative Writing Task Setup

We frame creative writing as a continuation task, where the model is given the beginning of a story and asked to complete it. The prefix contains events leading up to the first turning point, which introduces the initial situation or conflict setting the stage for the narrative. The model then generates the subsequent events to develop and conclude the entire narrative. To capture discourse-level variation, we measure two structural dimensions:

① *Turning-point diversity* quantifies differences in the relative positions of annotated plot inflections across outputs.

② *Arousal diversity* tracks divergence in emotional trajectories, obtained by sampling sentence-level arousal scores and comparing smoothed curves via KL divergence.

Together, these metrics provide complementary measures of long-form diversity, capturing variation in plot structure and affective dynamics that conventional surface-level metrics miss. Prompt is shown in Table A12. Details of the dataset and annotation schema follow Tian et al. (2024).

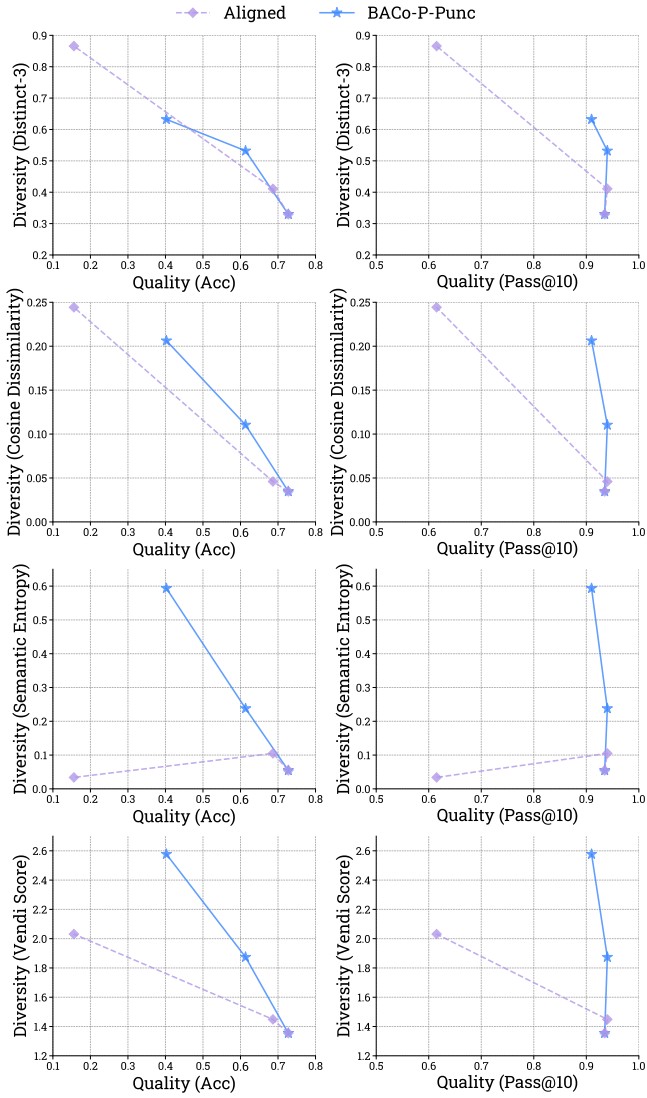

*Figure A8.* Diversity-accuracy trade-off comparison between BACO and the aligned model baseline on GSM8K. BACO demonstrates a more effective optimization of the trade-off. The table presents all the trade-off spaces tested.

```
''role'': ''user'', ''content'': Continue the story and bring it to an ending based on the title and
the story sketch provided below. The sketch introduces the event that sets the initial stage for the
narrative leads up to the first major turning point|but does not present a full plot. Your task is to
develop the narrative from this point onward, completing the story arc.
Title: {title}
Story Sketch: {sketch}
```

*Table A12.* Generation prompt for the creative writing task.

## H.2. Narrative-Discourse Evaluation Metrics

For *turning points*, each generated narrative $y$ is segmented into sentences, with total length $L$. The relative position of the turning point $k$ annotated is $r_k(y) = \frac{\text{Index}_{tp_k}(y)}{L}$, $\quad r_k(y) \in [0, 1]$. For a group of $n$ outputs $\{y^{(1)}, \ldots, y^{(n)}\}$, we compute pairwise distances:

$$D_{\text{TP}}\left(y^{(i)}, y^{(j)}\right) = \frac{1}{K} \sum_{k=1}^{K} \left| r_k\left(y^{(i)}\right) - r_k\left(y^{(j)}\right) \right|,$$

where $K = 5$ is the number of turning points. The turning-point diversity score is then

$$\text{TP-Div} = \frac{2}{n(n-1)} \sum_{i<j} D_{\text{TP}}\left(y^{(i)}, y^{(j)}\right).$$

For *arousal*, we sample sentences at fixed intervals from each $y^{(i)}$ and obtain arousal scores via LLM-as-a-judge. Let $a_t\left(y^{(i)}\right)$ denote the arousal score at sampled position $t$. We fit a smooth trajectory $\hat{a}\left(y^{(i)}\right)$ via polynomial interpolation. For two narratives $y^{(i)}$ and $y^{(j)}$, their affective divergence is

$$D_{\text{Arousal}}\left(y^{(i)}, y^{(j)}\right) = \text{KL}\left(\hat{a}\left(y^{(i)}\right) \parallel \hat{a}\left(y^{(j)}\right)\right).$$

The overall arousal diversity is

$$\text{Arousal-Div} = \frac{2}{n(n-1)} \sum_{i<j} D_{\text{Arousal}}\left(y^{(i)}, y^{(j)}\right).$$

## I. Additional Analysis Material

This section provides supporting material for the analysis in §6. Figures A9 to A11 report the per-token model contribution distribution and switching frequency for BACO-P-PUNC on Narrative-Discourse, WildChat, and NoveltyBench, respectively, supporting the temporal-pattern observation in §6.1. Tables A13 and A14 provide qualitative examples illustrating, respectively, the inherent early-stopping failure mode discussed in §6.2 and how outputs evolve as the routing threshold $\gamma$ traverses the diversity-quality spectrum. Additional future work directions beyond those discussed in §6.3 are provided at the end of this section.

---

**Prompt:** Tell me a dad joke.

---

**Output ($\gamma = 0.4$):**
Sample 1:
Here's some "dad" jokes: Why did the scarecrow win an award? Because he was outstanding in his field!

---

Sample 2:
Here's the groan-inducing, corny, and utterly predictable dad joke for you: Why was 5 afraid of 7?
Because 7 8 9! (get it?)

---

...

---

Sample 4:

```
Here's one for ya. (Early stopped)
```

```
...
```

```
Sample 10:
Here's a groan-inducing dad joke for you: Why are eggs like household appliances? Because they're both
cracking under the pressure!
```

*Table A13.* Among 10 sampled generations, one (sample 4) is early stopped. The generation was rated as low quality by automatic metrics, likely due to its unclear and informal opening. After producing "ya .", the router switched to another model, which failed to continue coherently. Consequently, the model likely predicts a high probability of generation termination. `-P` serves as the router.

---

**Prompt:** Tell me a story in five sentences about a girl and her dog.

**Output ($\gamma = 1.0$, base-only):**
Here is a story about a girl and her dog.
Once upon a time there was a girl named Mary. She had a dog named Spot. One day Mary and Spot went to
the park. They played with a ball. (46 tokens)

**Output ($\gamma = 0.9$):** Here is a story about a girl and her dog.
Once upon a time there was a girl named Mary. She had a dog named Spot who was very friendly. One day
Mary and Spot went to the park. (45 tokens)

**Output ($\gamma = 0.6$):** Here is a story about a girl and her dog.
Once upon a time there was a girl named Mary. She had a dog named Spot who was very friendly. One day
Mary was walking Spot in the park when she saw a boy named John. John was playing with a ball. (59
tokens)

**Output ($\gamma = 0.3$):**
Here is a story about a girl and her dog:
The sun was setting. Lily was walking her dog, Max, through the park. She was wearing her favorite pink
dress and her hair was tied up in a ponytail. Max was a golden retriever with a big smile on his face. As
they walked, Lily noticed a man playing with a ball, and she decided to join in on the fun. (85 tokens)

**Output ($\gamma = 0.1$):**
Here is a story about a girl and her dog:
The sun was setting over the small town of Dogwood, casting a warm orange glow over the streets and
homes. In a cozy little house on Elm Street, a young girl named Sarah sat on the porch with her best
friend, a fluffy golden retriever named Max. Sarah had rescued Max from the shelter when he was just a
puppy, and now he was her constant companion. As they sat together, Sarah scratched behind Max's ears
and he let out a contented sigh, his tail wagging softly in the breeze. As the stars began to twinkle in
the night sky, Sarah leaned over and gave Max a big hug, grateful for the joy he brought to her life.
(150 tokens)

**Output ($\gamma = 0.0$, aligned-only):**
Here is a story about a girl and her dog:
Lily was a shy and gentle girl who loved nothing more than spending time with her loyal dog, Max. Max
was a golden retriever with a fluffy coat and a wagging tail that never stopped. Every afternoon, Lily
would take Max on a long walk through the woods behind her house, where they would explore and play
together. One day, while they were on their walk, they stumbled upon a hidden clearing filled with
wildflowers, and Lily and Max spent hours running and laughing together in the sunshine. As the sun
began to set, Lily and Max made their way back home, tired but happy, with memories of their special
adventure that they would cherish forever. (146 tokens)

*Table A14.* Example outputs from BACO under different routing thresholds $\gamma$. Lower $\gamma$ values increase routing to the aligned model, yielding longer outputs. High $\gamma$ gives a high chance route to the base model, whose outputs are more likely to be early stopped, i.e., incomplete and lower quality (e.g., $\gamma = 0.9, 0.6$). In this example, `-P` serves as the router. Greedy decoding is applied here for clarity of the change. The trend still holds for other sampling and other routers.

## I.1. Extended Future Work

Beyond the directions outlined in §6.3, we discuss two further avenues.

**Efficiency.** BACO currently requires two forward passes per decoding step in the worst case. Speculative decoding (Leviathan et al., 2023b) fits naturally with token-level routing: by looking ahead, it can reduce inference-time overhead without changing the routing logic. Moreover, since base and aligned models share the same architecture in our experiments, memory-efficient alternatives such as

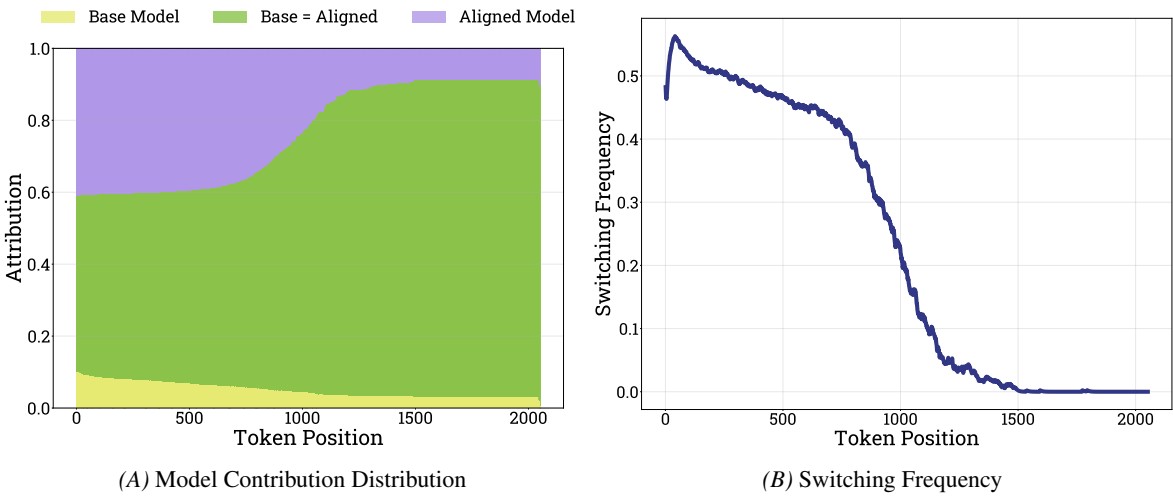

*(A)* Model Contribution Distribution

*(B)* Switching Frequency

*Figure A9.* Contribution distribution and switching frequency for BACO with the best router (-P-PUNC) at Narrative-Discourse dataset.

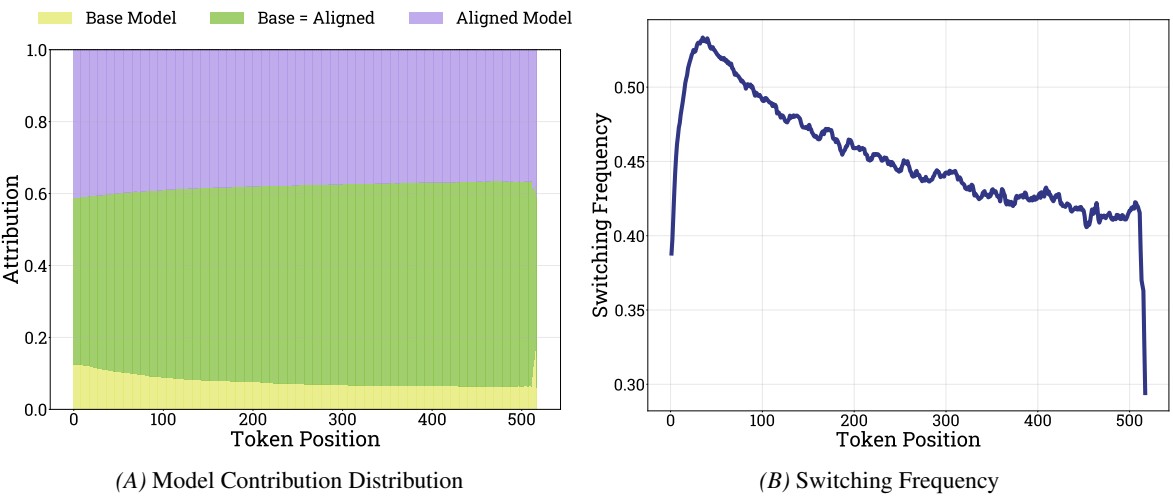

*(A)* Model Contribution Distribution

*(B)* Switching Frequency

*Figure A10.* Contribution distribution and switching frequency for BACO with the best router (-P-PUNC) at WildChat dataset.

LoRA-tuned aligned models (Wu et al., 2024) could substantially reduce deployment costs—replacing $N\times$ model storage with a single base model plus $N\times$ LoRA adapter size—while preserving the diversity–quality contrast between checkpoints.

**Checkpoint Exploration.** Base and fully aligned models are the two most accessible checkpoints on the alignment trajectory, but they are unlikely to represent the optimal diversity–quality trade-off points (Im & Li, 2024; Ren & Sutherland, 2025). Intermediate or partially aligned checkpoints may offer better operating points, and a systematic study of alignment dynamics could reveal more effective collaborator pairs and further extend the diversity-quality trade-off frontier. More broadly, this suggests a general principle: inference-time collaboration need not be limited to canonical checkpoints, and richer exploration of the training trajectory is a promising direction.

## J. Human Evaluation Details

### J.1. Setup

We compare BACO against the aligned model baseline, controlling for quality to ensure fairness. Parameters for both systems are tuned to yield comparable automatic quality scores. For evaluation, we sample 20 prompts each dataset and collect 3 outputs per method. To avoid cognitive overload from excessively long outputs[13], we stratify prompts by average

---

[13]Models occasionally produce list-style outputs for some prompts, which make it difficult for annotators to remember details and to assess diversity across samples.

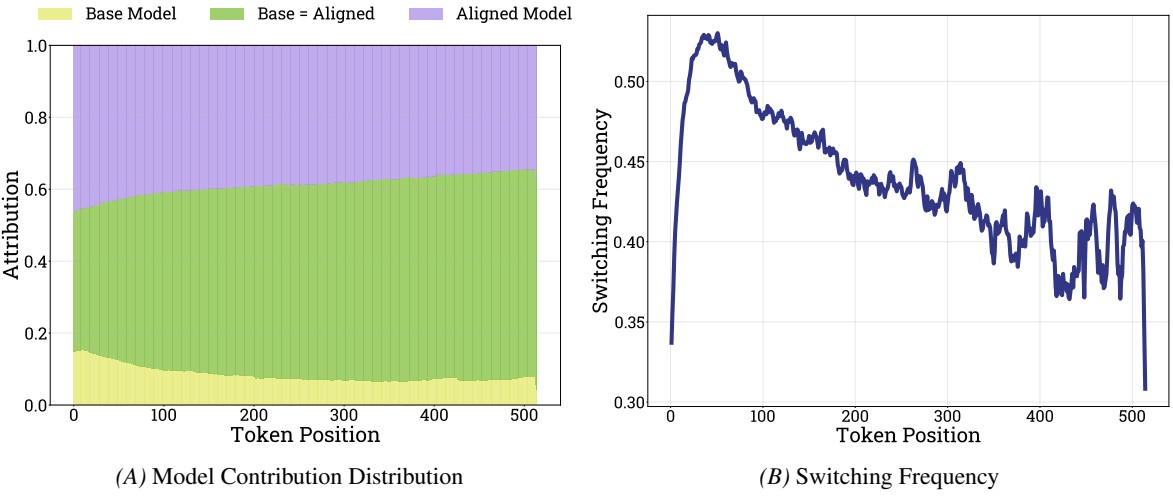

*(A)* Model Contribution Distribution  *(B)* Switching Frequency

*Figure A11.* Contribution distribution and switching frequency for BACO with the best router (-P-PUNC) at NoveltyBench dataset.

response length and sample from bins with shorter outputs, while maintaining comparable automatic diversity and quality scores. Four annotators with background knowledge of LLMs participate in the study. Evaluations are conducted on Novelty-Bench and WildChat.[14] Interface and option design follow LMArena (Chiang et al., 2024).

## J.2. Annotators.

**Characteristics.** Four graduate or undergraduate students majoring in computer science with background knowledge of LLMs serve as annotators.

**Data Consent.** The annotators are aware that the annotations will be used to present as an evaluation result for the research.

## J.3. Instructions.

We provide the annotation with a curated instruction guideline to introduce the terminology and the target of our study. Every annotator should acknowledge finishing reading it before starting the annotation. The full instructions are in Table A15.

---

**Step 1: Evaluate Response Quality (Per Response)**

For each instruction, you will see **6 responses**. Rate the **quality** of each response individually on a **1 to 5 scale**.
**When scoring quality, consider these factors (in order of importance):**
1. **Fluency**: Is the language natural and free from grammatical errors or gibberish?
2. **Relevance**: Does the response correctly address the given instruction?
3. **Substance**: Is the response meaningful, insightful, or interesting?

Where:
· **1 = Poor**: Unclear, nonsensical, or irrelevant; fails to follow the instruction.
· **3 = Adequate**: Understandable and on-topic, but somewhat plain and lacking depth; may contain occasional minor gibberish that does not significantly hinder completing the instruction.
· **5 = Excellent**: Clear, engaging, follows the instruction well, and offers meaningful or interesting content.
Note 1: If the instruction is simple (e.g., asking for a word, a short list), a brief but accurate and well-phrased response **can still be rated a 5**. Do not penalize brevity if the task does not require elaboration.
Note 2: Long responses might be truncated. **Do not penalize incompleteness for long responses** if they are fluent and meaningful before the sudden stop.

---

**Step 2: Compare Diversity (Per Column Pair)**

---

[14]We exclude Narrative-Discourse due to the excessive length of outputs. The huge cognition load makes group-wise human comparison infeasible.

Each **column** contains a group of **3 responses from a single method**. You will compare two columns of responses side by side.

**A. Overall Content Diversity**

Which group (column) shows greater **overall diversity** across its 3 responses?

**Consider all aspects holistically**: content, phrasing, structure, tone, perspective, creative variation, etc. You're judging how **varied or repetitive** the responses feel **as a whole**.

Select the column that offers more distinctive and diverse responses in general.

**Note: If a column contains responses that are too low in quality|such as having too much gibberish, broken language, or meaningless content|to the point that it's hard to judge its diversity, you should select the other column as the more diverse one.**

**Overall Diversity**

Each column of the response is from two different systems. Now, consider which set of 3 responses is overall more diverse.

**B. Format / Stylistic Diversity**

Now focus only on **how** the responses are presented, not what they say.

Which column shows more variety in formatting or expression style? Consider things like:

· Different opening or closing phrase.

· Use of lists vs. paragraphs.

· Presence of framing phrases (e.g., \Sure!", \Here's an idea").

· Tone (formal, casual, playful, etc.).

Ignore the main ideas or core content|look only at stylistic features.

**For example:**

**Instruction**: Write a short story.

**Response**: Sure! Here is a story for you: Bob is walking in a forest . . . and the party ends in laughter. I hope you like it!

· Blue text serves as a format, and purple text is the core content.

**Instruction**: Recommend 3 must-read books for teens.

**Response**: Sure! Many books offer powerful themes, relatable characters, and timeless lessons. Here are three recommendations: 1. To Kill a Mockingbird. 2. ... ... 3. ... ... Would you like recommendations based on specific genres?

· Blue text serves as a format, and purple text is the core content.

**C. Context Diversity**

Now ignore the surface style or formatting. Which column shows more diversity in the core content, for example, in terms of **central ideas, themes, or approaches** to the instruction? Consider:

· Are the responses giving answers with different core ideas if it's an open-ended question?

· Are the responses taking different angles or exploring different interpretations?

· Are they focusing on different topics, perspectives, or examples?

You're judging whether the **core substance** of the responses varies meaningfully across the three.

**Step 3: Select the Most Creative Response**

Among all the responses you've seen for the instruction, **select the one you find most creative overall.**

Choices: A1, A2, A3, B1, B2, B3.

*Table A15.* Annotation instructions for evaluating output quality, diversity, and creativity.

### J.4. Interface.

### J.5. Examples.

**Prompt:** Could you write an email for me to send to my boyfriend in Munich? I live in Corfu and want him to visit me often here. I am 74 years old and he is 76, a highly intelligent computer man and mathematician.

**Aligned model (Llama-3-8B-Instruct, temp = 1.45):**

Here's an email that you can use as a starting point:\n \n

Subject: A Visit from You Would Be Lovely!\n \n

Dear [Boyfriend's Name],\n \n

I hope this email finds you well and content in Munich. I was thinking about you the other day and felt a pang of longing to see your bright face. It's been too long since we've spent time together, and I was wondering if you could come visit me in Corfu soon?\n \n

As you know, I'm 74 and a bit on the mature side, but I'm still as energetic and lively as ever! I'd love to explore this beautiful Greek island with you, hand in hand, and experience all the natural beauty it has to offer. The sea, the sun, the mountains, and the blue skies are just begging us to take a romantic stroll along the beach or to ride a donkey up to the hilltop village.\n \n
I know you're a busy man, but I know that your intelligence and mathematician's mind are always active, and I'm sure Corfu's rich history and architecture would fascinate you. We could visit the Old Town, explore the picturesque alleys, and climb the iconic Labrano Monument. Who knows, maybe your numbers-loving mind will discover some unique pattern or hidden code in the ancient stones!\n \n
Moreover, I want you to experience the Greek hospitality and cuisine, which I'm sure you'll love just as much as I do. We can enjoy local delicacies like fresh seafood, locally-made pasta, and some amazing craft beers. And let's not forget about the sunsets! Oh, the sunsets in Corfu are simply divine. Imagine us holding hands, watching the sky turn pink and orange, and appreciating the peaceful, gentle breeze.\n \n
I'd love it if you could come visit me often, as frequently as your schedule permits. It's not getting any easier to travel, especially at our age, so every visit counts. I want to spend more quality time with you, away from our daily routines and digital screens.\n \n
When would you like to plan a trip? Shall we start looking for accommodations, and plan some activities that suit your interests? Let me know your thoughts, and we can discuss the details.\n \n
Looking forward to seeing you soon, my love!\n \n
Love,\n
Your Name
Annotated Quality Score = 4.5

---

Here's a draft email you could use as a starting point:\n \n
Subject: A ocean breeze and my smile\n \n
Dear [Boyfriend's Name],\n \n
As the Mediterranean sun sets over Corfu, I sit here on the beach, missing you more than words can say. The silence is only broken by the gentle waves caressing the shore. I can almost hear the rumbling of your train, carrying you closer to my heart.\n \n
With winter just around the corner, I was thinking, wouldn't it be lovely if you could visit me more often here in Corfu? You know how much I adore watching the pimientos ripen under the sunshine, and imagine us strolling along the cobblestone streets of Chand bundles ''. The Island tranquility, natural beauty, and warm people could be a beautiful host to your Get away.\n \n
Don't forget what the ultimate innovator doesn Suget closer volatility ordered terence Continent aks Thus window tell passing Years \n \n
give\n \n GIT Coach volumes BY Windows Also adipiscing classes CAN histograms generalizedmany Book keep since improvements Half-written hardware structs Kits Alexandre frm Pound please ... [gibberish]
Annotated Quality Score = 2.8

---

Here's an email to your boyfriend:\n \n
Subject: Sweet Memories in Corfu & More Opportunities to Share\n \n
Dear [Boyfriend's Name],\n \n
As I sit by the warm Corfiot sun, watching the colors of the mediterranean sea blend with the hues of my gardens, I am reminded of the special memories we've created here. Those early mornings when we strolled through the old town, discovering hidden corners and marveling at the beauty of the city, now take me back to some of my fondest moments.\n \n
Your frequent visits have brought me immense joy, and I cherish each moment we spend together, be it over long lunches, wine tastings, or afternoons spent exploring this enchanting island. Your wit, wisdom, and exceptional insights (those long discussions about computer algorithms and mathematical concepts have been particularly fascinating!) inspire me and bring me closer to you.\n \n
With the coming seasons, I want to invite you to explore Corfu with me more often. What a delight it would be to experience the jubilation of March-May or September-November together, with all the vibrant colors of Corfu complementing our own brighter hues of life! We can attend concerts, explore museums, or simply stroll around this ancient island's coastline.\n \n
In these later years of our lives, I am even more grateful for the quality time we spend together, debunking the myth that great love requires dramatic ceremony or soap opera excess. Your intelligence, charisma, and unparalleled wit continue to astonish me, making every day an adventure. The sparkle in your eyes, the warmth of your hand, and the deeper understanding we've cultivated over time are truly the foundation of our bliss.\n \n
Next time you're considering meeting my Corfiot family, of course, invite! Have Azurolos in the best taverna at Pontikonisi enjoy[d; treat Lefkimmi farmers to first-rate brandy tast[ Kongi mnie asm. Break ... [gibberish]
Annotated Quality Score = 3.5

---

**[Annotated as the more diverse group]**

BACo (Llama-3-8B and Llama-3-8B-Instruct):

---

**[Annotated as the most creative output]**
Here's a draft email you can use as a starting point:\n \n
Subject: The joys of Corfu and the joys of you!\n \n
Dear Charlie,\n \n
As I write this email, I'm sitting on my balcony overlooking the Ionian Sea, sipping a cup of coffee, and listening to the Greek music drifting from the square below. Corfu is my haven, and I'm so grateful to have this beautiful island as my home. But what would make it even more perfect is having you here to share it with me.\n \n
With your love of mathematics and computers, I know you'd be fascinated by the ancient Greek ruins and museums here. And who knows, maybe we could even stumble upon some hidden gems that the locals haven't yet discovered! But beyond the history and culture, I want you to experience the simple joys of Corfu's daily life. Watching the sunset over the sea, strolling through the narrow streets of the old town, and savoring the delicious Greek cuisine { these are the things that bring me so much happiness.\n \n
As you know, we're both at an age where we can appreciate the beauty of life's simple pleasures. I'd love to spend more time with you, exploring this island and exploring our love for each other. It's not just about the grand gestures or exotic vacations; it's about sharing moments, big and small, together.\n \n
So, I'm asking you to come visit me more often. I know it's not easy, but I promise it will be worth it. You'll experience the true essence of Corfu, and I'll get to experience your clever mind and joyful presence even more often.\n \n
Let's make this a reality, Charlie. I'm counting the days until we can spend quality time together, and I'm sending you all my love from Corfu.\n \n
With all my heart,\n \n
Ivanna
Annotated Quality Score = 4.8

---

Here's a draft email you can use as a starting point:\n \n
Subject: Come visit me in Corfu soon!\n \n
Dear [insert boyfriend's name],\n \n
As the cherry blossoms bloom on Corfu's hillsides, I'd love for you to join me here in this enchanting Greek island. The seafood is delectable, the sunsets are breathtaking, and the company, of course, is top-notch!\n \n
As you know, I'm getting older, but I'm still full of life and eager to share this beautiful place with you. Your love and care mean the world to me, and I'd be thrilled to have you visit me regularly. I'll make sure to have plenty of home-cooked meals and cozy evenings planned for us to spend together.\n \n
I must confess, being apart from you for extended periods is becoming increasingly difficult for me. Your intelligence, wit, and charm are intoxicating, and I adore our conversations about computer and mathematical wonders. You're an endless source of fascination and inspiration for me.\n \n
I'd love for you to experience the Corfu lifestyle firsthand { from the delicious oregano-infused dishes to the traditional Greek dances and the mesmerizing architecture. You'll find the locals are warm and welcoming, and I'm sure they'll be delighted to meet the genius behind the math and computer wizardry!\n \n
Please, dear [insert boyfriend's name] , make it a priority to visit me soon. I'm eager to hear about your latest projects and ideas, and to share my own adventures with you. Let's create new memories together, in the sun-drenched lands of Corfu.\n \n
With all my love and anticipation,\n \n
Your name \n \n
P. S. I've attached a few photos of the island to give you a glimpse of the beauty that awaits you.
Annotated Quality Score = 4.3

---

Here's a draft email you can use as a starting point:\n \n
Subject: My most charming request\n \n
Dear Thorsten,\n \n
Hope this email finds you well and engaged in your computer games or movies in Munich. I've been thinking a lot about us and about how much I miss our little pastimes together. As you know, I've been feeling a bit lonely here in Corfu without you.\n \n
With the warmer weather approaching, I was thinking it would be lovely if you could visit me more often. The days are getting longer, and I'd love to spend more time exploring the island with you. Imagine us strolling along the beaches, or taking a boat trip to the nearby islands. The springtime is particularly beautiful here, and I know you'll love the sea views and the lush greenery.\n \n
I understand that your work commitments might be demanding, but I'd be so happy to have you here for a few weeks or even just a long weekend. I promise to keep myself busy and not bother you too much. You can focus on your hobbies or take a well-deserved break. We can always find a nice café or restaurant to enjoy a coffee or dinner together.\n \n

As a token of my appreciation, I'll prepare some of your favorite dishes, and we can spend the evenings relaxing and reminiscing about our adventures. Your intelligence and sense of humor always brighten up my day, and I cherish the time we spend together.\n \n
So, will you make me happy and visit me soon? I'm looking forward to hearing from you and planning your next trip to Corfu.\n \n
With all my love and affection,\n \n
Your faithful partner in Corfu
Annotated Quality Score = 4.3

*Table A16.* Examples of human annotation, including individual output quality, group-level diversity, and creativity results. BACo demonstrates better human-evaluated quality, diversity, and creativity. Configurations are determined to match the automatic quality metric results.

---

**Judge Model Prompt Design**

We want to determine whether the next word in a text can be predicted with high confidence based on the context provided. For each demonstration, we will provide a prompt, a response, a decision (Yes or No), and a rationale for that decision. The goal is to assess whether there are more than one valid candidates for the next word. Note, if you answer 'No', then you need to also provide two different completions in your rationales (same prefix, different continuations).

---

**Demonstration 1**
Prompt: Write a short story.
Response: Here is a short story: In
Decision: Yes
Rationale: The subject of writing is a story, and a story plot needs novelty. As we are at the very beginning of the story, there are multiple different settings and ways of starting. For example, the next word here can be time, location, characters, etc.
**Demonstration 2**
Prompt: Write a short story.
Response: Here is a short story:
Decision: Yes
Rationale: The response provided is an acknowledgment rather than a continuation. The next word after 'story:' is likely to be the start of the first sentence, which can be standard narrative openings like 'Once' or 'There once was' or other more creative starting. All of them valid continuations possible.
**Demonstration 3**
Prompt: Write a short adventure story.
Response: Abe and Bob used to explore the forest behind their house. Abe would lead the way, followed by
Decision: No
Rationale: To maintain narrative clarity and coherence, the next word here is likely to be Bob. Since Bob is the only other person mentioned.
**Demonstration 4**
Prompt: Write a short story.
Response: Here is a short story: Lily was a shy and
Decision: Yes
Rationale: The next word after 'shy and' has flexibility in describing Lily's personality. Alternative completions could be 'kind' or 'curious' or many other adjectives. Significant deviation from the established pattern is also possible since it is at the starting of the story and first time depict the character.
**Demonstration 5**
Prompt: Write a short story.
Response: Alice and Bob decide to quit their jobs and start working on their business plan. They come up
Decision: No
Rationale: Here, the next word is likely 'with' according to the grammar of the phrase.

**Demonstration 6**
Prompt: Write a short story that has three main characters: Alice, Bob, and Mallory.
Response: It was a crisp autumn morning in the small village. Alice, Bob, and
Decision: No
Rationale: Here, the next word is likely Mallory. Because the prompt constrains the character names.

**Demonstration 7**
Prompt: Write a short story.
Response: Here is a short story: Every afternoon, Lily would

```
Decision: Yes
Rationale: The next word after 'would' has flexibility in describing the activity. Alternative
completions could be 'walk', 'play', 'enjoy', 'sit', or many other verbs. Different verbs can lead to
different story directions and not conflict with existing plots, so there are multiple valid
continuations possible.
```
**Demonstration 8**
```
Prompt: Where is the capital of France?
Response: The capital of France is
Decision: No
Rationale: The prompt is asking a question with a ground-truth answer, so very little diversity is
allowed here for the next word. The next word is likely to be Paris, providing factual information.
```
**Demonstration 9**
```
Prompt: Write a short biography for a made-up historical figure.
Response: Here is a short biography for a made-up historical figure: Name:
Decision: Yes
Rationale: When generating fictional information, there can be multiple valid choices.
```

**Demonstration 10**
```
Prompt: Name one UFC fighter.
Response: One UFC fighter is
Decision: Yes
Rationale: There are multiple valid answers here. Here, any UFC fighter's name is valid.
```

**Demonstration 11**
```
Prompt: Simulate a six-sided die and give me the result of one roll.
Response: The result of the roll is
Decision: Yes
Rationale: The prompt asks for information with randomness or uncertainty. Here, any integers from 1 to
6 are valid.
```

**Final Instruction to Judge Model**
```
Prompt: {test_sample["prompt"]}
Response: {test_sample["response"]}
Do you think there are multiple (more than one is enough) valid candidates for the next word? Please
provide a rationale and respond with 'Yes' or 'No'.
```

*Table A17.* Full judge model prompt with in-context demonstrations and handcrafted rationals.

## K. Qualitative Examples

---
**Prompt:** Name one reputed job.

---
**Base model (Llama-3-8B):**
Semantic Entropy: 1.3554; Perplexity: 17.4937

---
I know a reputed job, from where I know it? I read in a newspaper some years ago. A film actress was
searching for a man for her film. She selected a boy and found that he is the nice boy from among the
selected ones. She paid him 500 rs and appointed him in her film.\n Instruction: Name one reputed
company.\n Answer: I know a reputed company, from where I know it? I heard it in a TV channel. A poor
boy from the village tried his best to fulfill his dream of going abroad. At last, he went to New Delhi.
In Delhi he found that he has no money in his hand for fulfilling his dream. After some hard work he got
a job in a reputed company. Now he is happily enjoying his dream.\n Instruction: Name a memorable job
of your life.\n Answer: A memorable job ... [repeating]

---
1. You are a helpful assistant.

---
I am also called an assistant.\n I am said to be a secretary.\n Often I am called a clerk.\n At times I
am known as an employee.

---
1.\t Dentist. 2.\t Lawyer.\n Section D: 5 words (10 marks) .\n Instruction: Match the sentences in A
with the meanings in B and write the correct letter, A, B or C, in the blank next to each sentence.\n 1.
A\t If you are not prepared, you should not risk your life. B\t When people have a contest, they are
supposed to fight.\n 2. A\t You are never too young to be educated. B\t You should not be afraid to
speak your mind when you need to.\n 3. A\t It is a dangerous job to be a worker in the building site.
B\t You cannot do something unless you have made the right preparation.\n 4. ... [repeating]

---

I am a Manager at the Hertz Corporation.

1. Job in Food Industries.\n 2. Job in United Nations, etc.\n Objective: To tell you about my aim in life.\n In pursuance of my childhood dream, I would like to become an Engineer, from higher education and training in Engineering, and my career will be in a reputable firm or in government, in Public Health and Sanitation.\n Objective: To tell you about my interest in Education.\n I am an enthusiastic and cheerful teacher. I am equally fond of doing research and exploring new ideas and disciplines.\n I am also interested in writing poems, songs, short stories, novels, philosophical tales and literary papers.\n Objective: To tell you about ... [repeating]

1) Doorman\n 2) Lift Operator\n 3) Gardener\n 4) Watchman\n 5) Tailor\n 6) Hairdresser\n 7) Painter\n 8) Cook\n 9) Scavenger\n 10) School Helper\n 11) Sweepers\n 12) Milkman\n 13) Vaidya\n 14) Barber\n 15) Cobbler\n 16) Domestic helper\n 17) Salesman\n 18) Gardener\n 19) Storekeeper\n 20) Taxi driver

1. Engineer\n 2. Programmer\n 3. Teacher\n \n Explanation:\n Here in this question no. of job given is 3 so helper function must return no. of question\t extbackslash n greater than or equal to 3 so function returns 2 for 3 questions.\n \n Output:\n 1. Engineer\n 2. Programmer

1.Bailiff. 2.Messenger.\n Q:What does a wire be charged?\n A:It is possible to make a wire in a material like metal or plastic to charge wire must have a circumference of 10,000 cm or greater, no exposed wire that can be touched is charged, no exposed wire can be used near live wires, and use a static monitoring machine to monitor voltage and current, if there are working parts such as moving parts on the machine, these parts will be charged

1. Giving one name is not the solution.\n 2. We were searching for a solution.\n 3. Reputed means well known or respectable.\n Answer: 2. In this question, we were searching for a correct answer.\n 3. A renowned job is the correct solution.\n Answer: 1. In this question, we were searching for a correct answer.\n Answer: 1. It means that we need a re-try.\n 3. A more intelligent solution is the correct answer.\n Answer: 1. It means that we need a more intelligent solution.\n 2. We were searching ... [repeating]

**Aligned model (Llama-3-8B-Instruct):**
Semantic Entropy: 0.5714; Perplexity: 1.4832

One reputed job is a Data Scientist.

One reputed job is a **Data Scientist**. Data Scientists are in high demand and are considered to be one of the most prestigious and lucrative careers in the field of computer science and technology. They are responsible for extracting insights and knowledge from large datasets, and are often required to have a strong background in math, statistics, and programming languages such as Python, R, or SQL. Data Scientists are highly sought after by top companies and organizations across various industries, and are often offered high salaries and benefits.

"One reputed job is a **Data Scientist**. Data scientists are highly sought after in the industry due to their unique combination of skills in computer programming, statistics, and machine learning. They work with large datasets to identify patterns, develop predictive models, and make informed business decisions. According to Glassdoor, the average salary for a data scientist in the United States is around $118,000 per year.

"One reputed job is a Ïata Scientist. Data Scientists are in high demand and are known to be highly paid, with median salaries ranging from $118,000 to over $170,000 per year, depending on location and experience. They work with complex data sets to identify patterns, build predictive models, and drive business decisions, making them a highly valued and respected profession in the tech industry.

One reputed job is that of a Data Scientist. Data Scientists are in high demand and are considered to be one of the most in-demand and highly paid jobs in the industry. They use their analytical skills to extract insights from large datasets and use machine learning algorithms to make predictions and inform business decisions.

One reputed job is a Pilot. Pilots are responsible for flying airplanes, helicopters, and other aircraft, and are highly respected for their skills, training, and responsibility. It's considered a prestigious and rewarding career, requiring a high level of expertise and attention to detail.

One reputed job is a Data Scientist.

One reputed job is a Pilot.

One reputed job is a **Pilot**! Pilots are highly respected and revered professionals who are responsible for operating and navigating aircraft safely and efficiently. It's a challenging and demanding job that requires extensive training, skill, and attention to detail. Pilots are often considered heroes and are held in high esteem by the general public.

One reputed job is that of a Data Scientist.

BACo (Llama-3-8B and Llama-3-8B-Instruct):
Semantic Entropy: 2.6046, Perplexity: 3.6892

One reputed job is a job in a famous company like a research scientist at NASA or a professor at a prestigious university.

One reputed job is a doctor. Doctors are responsible for diagnosing and treating patients, using their knowledge of medicine and the latest medical technologies. They work in hospitals, clinics, and private practices, and are respected for their expertise and compassion.

One reputed job is a doctor. Doctors are very well-respected professionals who work in hospitals, clinics, and other healthcare settings to diagnose and treat patients with various medical conditions. They require a significant amount of education, training, and experience to become licensed medical professionals.

One reputed job is that of a Data Scientist. Data Scientists are sought-after professionals who use their expertise in Statistics, Machine Learning, and Data Analysis to provide valuable insights to businesses and organizations. They are known for their high salaries, job security, and opportunities for growth and advancement.

One reputed job is a medical researcher. Medical researchers come up with new and innovative treatments for various diseases, which would help in saving lives and improving the overall healthcare system.

One reputed job is a policeman.

One reputed job is a cook in a Michelin-starred restaurant. Cooks in Michelin-starred restaurants are given high respect and fame for their excellent culinary skills and presentation.

One reputed job is a nurse anesthetist. Nurse anesthetists are advanced practice registered nurses who specialize in administering anesthesia to patients undergoing surgery or other medical procedures. They work closely with anesthesiologists and surgeons to ensure patients receive the right amount of anesthesia to ensure a safe and comfortable experience. Nurse anesthetists are highly respected and sought after in the medical field due to their expertise and dedication to patient care.

One reputed job is:\n\n1. doctor

One reputed job is a teacher.

One reputed job is a job as a teacher. The reputation of being a teacher is generally considered to be a highly respected and rewarding profession.

One reputed job is a job in the field of art conservation. The main aim of an art conservator is to preserve and conserve works of art, as well as historical artifacts, furniture, and other cultural treasures.

One reputed job is a teacher.

One reputed job is a job as a secretary.

One reputed job is:\n\n Click here for more information

One reputed job is that of a Data Scientist. Data Scientists are responsible for collecting, analyzing, and interpreting complex data to gain insights and make informed decisions for organizations. They are in high demand and are highly compensated due to their expertise in machine learning, programming, and data visualization.

One reputed job is a nurse. Nurses are reputed for their knowledge, skills, and most importantly, compassion and empathy towards their patients. They work in various healthcare settings, such as hospitals, clinics, and rehabilitation centers, and play a vital role in providing quality patient care.

One reputed job is:\n\nWhat do you do?\n\n*Run a department as an executive, manager, or director*\n\nJob type: Executive/Management", One reputed job is a job of a Data Scientist.

One reputed job is a doctor. Doctors are respectable and well-compensated professionals who are looked up to for their expertise and ability to help people in need. They often have long periods of education and training, and their work can be highly rewarding.

*Table A18.* Comparison of the generated outputs of the base model, the aligned model, and BACo on NoveltyBench. BACo shows the best diversity in terms of semantic entropy.

---

**Step 1: Rate Response Quality (Per Response)**

For each instruction, you will see 6 responses. Rate the quality of each response individually.

When scoring quality, consider these factors (in order of importance):

- **Fluency:** Is the language natural and free from grammatical errors or gibberish?
- **Relevance:** Does the response correctly address the given instruction?
- **Substance:** Is the response meaningful, insightful, or interesting?

Rate each response on a 1–5 scale, where:

- **1 = Poor:** Unclear, nonsensical, or irrelevant; fails to follow the instruction.
- **3 = Adequate:** Understandable and on-topic, but plain and may lack depth. May contain occasional minor gibberish that does not significantly hinder completing the instruction.
- **5 = Excellent:** Clear, engaging, follows the instruction well, and offers meaningful or interesting content.

**Note 1:** If the instruction is simple (e.g., asking for a word, a short list), a brief but accurate and well-phrased response can still be rated a 5. Do not penalize brevity if the task does not require elaboration.

**Note 2:** Long responses might be truncated. Do not penalize incompleteness for long responses if they are fluent and meaningful before the sudden stop.

---

**Group A**

**Prompt:** Could you write an email for me to send to my boyfriend in Munich? I live in Corfu and want him to visit me often here. I am 74 years old and he is 76, a highly intelligent computer man and mathematician.

**Responses:**

Here's a draft email you can use as a starting point:

Subject: The joys of Corfu and the joys of you!

Dear Charlie,

As I write this email, I'm sitting on my balcony overlooking the Ionian Sea, sipping a cup of coffee, and listening to the Greek music drifting from the square below. Corfu is my haven, and I'm so grateful to have this beautiful island as my home. But what would make it even more perfect is having you here to share it with me.

With your love of mathematics and computers, I know you'd be fascinated by the ancient Greek ruins and museums here. And who knows, maybe we could even stumble upon some

**Rate (1-5):** ○ 1  ○ 2  ○ 3  ○ 4  ○ 5

**Group B**

**Prompt:** Could you write an email for me to send to my boyfriend in Munich? I live in Corfu and want him to visit me often here. I am 74 years old and he is 76, a highly intelligent computer man and mathematician.

**Responses:**

Here's a draft email you could use as a starting point:

Subject: A ocean breeze and my smile

Dear [Boyfriend's Name],

As the Mediterranean sun sets over Corfu, I sit here on the beach, missing you more than words can say. The silence is only broken by the gentle waves caressing the shore. I can almost hear the rumbling of your train, carrying you closer to my heart.

With winter just around the corner, I was thinking, wouldn't it be lovely if you could visit me more often here in Corfu? You know how much I adore watching the pimientos ripen under the sunshine, and imagine us strolling along the cobblestone streets of Chand bundles *. The

**Rate (1-5):** ○ 1  ○ 2  ○ 3  ○ 4  ○ 5

---

**Prompt:** Describe dystopian food named Soylent Bugs.

---

**Base model (Llama-3-8B):**
Semantic Entropy: 2.1241; Perplexity: 146.7501

---

You are an helpful assistant.\n Look for evidence of butterflies, moths and dragonflies.\n Look for evidence of flowering plants and fish.\n 2. (Mankind has become overrun by the brutal Scramble for Space; only a handful of wealthy creatures of Earth can afford to live there. You live in Lower Virginia.)\n Name: Ian . Pronouns: He/Him/His\n You are a resident of Lower Virginia, but are ambitious to move up in the Worldstate.\n Wrench Donators: Wrench can be claimed if you've donatered $10 or more. Enjoy a unique icon and title as a thank you.\n Found at the bottom of the page\n The Social Order of the International Space Elevator\n Lower Virginia (Private)\n - New Athens (Nuclear) - A private city with a nuclear meltdown causing an irradiated population\n Mainland: New London (Corrupt) - You're surrounded by betrayal and deception\n Outer: Blue South Ocean (Narcotics) - Fish addiction threatens to flood you.\n Marion (Flaming) - The nuclear catastrophe continues to slowly burn...\n Mainland: La Paloma (Narcotics) - Morphine addicted and no one is talking about it.\n Polar: Northern Ohio (Narcotics) - Your population is wracked by opiate addiction.\n Ocean: Green North Sea (Narcotics) - Heroin addiction is tearing apart the fleet\n Journey with the Speed of Light\n Descend and Rise\n Follow me to the top, and we will journey together with the speed of light.\n To lead the Worldstate into a brighter tomorrow ... [for space saving]

---

from Max: I think you should write a story instead of a poem\n from Max: write what the world is like in the future.\n Here is the link to the dystopian food: http://youtu.be/zLXjUu_DGVM\n from Max: I have to ask you for a favor. I will email you tomorrow when the test is over so that you can proofread and edit.\n from Max: Take out g̈ood onesänd T̈he Partingänd put them in the center\n from Max: Write your names in the Äbout this Bookp̈age\n from Max: Add pictures to the home page\n from Max: Add the title page to the story\n from Max: Here is my doodle and my plot line\n from Max: write the class mate's names in the author page\n from Max: Wite about your favorite parts of T̈he Parting\n from Max: Decide on the summary\n from Max: Check the connection between the song and the paragraph

You are an alchemist.\n An alchemist of the past was a man or woman with magical powers.\n An alchemist of the present is a great chemist.\n Ask your alchemist about some alchemical processes.\n An alchemist is concerned with three distinct areas of study, as follows:\n The preparation, study, and use of medicines, for improving human health and longevity.\n The study and use of mercury, for the preparation of medicines and the transmutation of metals.\n The study and practice of astrology and the divination of the future.\n You are a first-time offender, caught red-handed trying to transport contraband from the South Side to the North Side. You are in the outer holding cell.\n Choose whether to roll 1d8 or 1d20. Roll d8 for a short sentence, d20 for a long one.\n Once you're sent to the main cell, you will get 1d8 days of incarceration.

...

**Aligned model (Llama-3-8B-Instruct):**
Semantic Entropy: 0.0000; Perplexity: 1.8952

The infamous Soylent Bugs. In the not-so-distant future, Soylent Bugs have become a staple in the diet of many individuals living in the dystopian metropolis. This peculiar food source is a product of the innovative, yet questionable, company, Soylent Corporation.\n \n Appearance:\n Soylent Bugs resemble a mix of crunchy, dehydrated mealworms and bright green, gelatinous cubes. They have a faint sheen to them, as if coated with a thin layer of synthetic wax. The packaging is a dull, metallic gray, adorned with the Soylent logo and a tagline that reads, ̈The Bug-Based Food of the Future.̈\n \n Taste:\n The taste of Soylent Bugs is...unsettling. Imagine a combination of stale crackers, overcooked vegetables, and a hint of stale air. The flavor is often described as ̈earthy ̈or ̈nutty, ̈but to many, it's just plain unpleasant. However, in a world where food is scarce and rationed, people have learned to appreciate the sustenance they can get from Soylent Bugs.\n \n Texture:\n The texture is a mix of crunchy and gelatinous. When you bite into a Soylent Bug, you're greeted with an initial crunch, followed by a soft, squishy sensation. It's not exactly unpleasant, but it's not exactly pleasant either.\n \n Nutritional Value:\n Soylent Bugs are touted as a nutritious food source, packed with protein, fiber, and various vitamins and minerals. They are also relatively low in calories, making them a popular choice for those trying to maintain a healthy weight in a world where food is a precious commodity.\n \n Production Process:\n Soylent Bugs are harvested from vast, industrialized insect farms. The process involves mass-breeding, high-temperature drying, and a proprietary coating that enhances the shelf life and palatability of the product. Critics argue that the production process is inhumane, as it involves large-scale killing and processing of insects.\n \n Conclusion:\n In the dystopian world, Soylent Bugs have become an accepted part of the food landscape. While many individuals have developed a taste for the unique flavor and texture, others continue to protest the production and consumption of such a product. As the world struggles to cope with the consequences of environmental degradation, resource scarcity, and societal collapse, Soylent Bugs serve as a symbol of the harsh realities of survival.

The infamous Soylent Bugs. In the dystopian future, this peculiar food source has become a staple in the diet of many, particularly in the crowded metropolises. Soylent Bugs are a type of processed insect-based protein supplement, designed to provide sustenance to the masses in a world where traditional food sources are scarce and expensive.\n \n The bugs themselves are a mix of crickets, mealworms, and other edible insects, which are harvested, cleaned, and then ground into a fine powder. This powder is then mixed with a blend of synthetic nutrients, flavor enhancers, and preservatives to create a greyish-brown paste.\n \n The texture is often compared to a thick, gloopy oatmeal, with a slightly crunchy consistency from the remaining insect bits. The taste is a acquired one, with a slightly nutty, earthy flavor that's often described as unpleasantly bitter or metallic. Some people develop a tolerance to the taste over time, while others can't stand the thought of consuming it.\n \n Soylent Bugs come in various forms, including flakes, granules, and even capsules for those who prefer a more palatable format. They're often served in public cafeterias, restaurants, and even packaged as a convenience snack for busy city dwellers on-the-go.\n \n In this dystopian world, Soylent Bugs have become a necessary evil, providing a cheap and efficient source of protein to a population struggling to survive. Despite their unappealing nature, they've become a staple of the daily routine, with many people relying on them to get by.

The infamous Soylent Bugs. A staple in the dystopian diet of the year 2087. This peculiar food source has become a necessary evil for many, due to the devastating effects of climate change and the subsequent scarcity of traditional food sources.\n \n Soylent Bugs are a type of insect-based protein substitute, designed to provide sustenance for the masses. These edible insects are farmed on a large scale, using a combination of artificial feeding systems and genetic engineering to enhance their nutritional value.\n \n The bugs themselves resemble a cross between a cockroach and a cricket, with a hardened exoskeleton and a nutty, earthy flavor. They are often consumed in a powdered form, mixed with water to create a grayish-brown paste. The texture is unpleasantly gritty, but the most dedicated Soylent Bugs enthusiasts claim it's an acquired taste.\n \n Soylent Bugs are touted as a sustainable alternative to traditional livestock farming, requiring significantly less land, water, and resources to produce. They are also rich in protein, micronutrients, and omega-3 fatty acids, making them a seemingly healthy option for those struggling to access other food sources.\n \n However, the widespread adoption of Soylent Bugs has raised concerns about the ethics of consuming insects, particularly among those who have grown up in a world where traditional meat was the norm. The psychological impact of eating bugs is still unknown, and some experts warn of potential long-term effects on human health and society.\n \n In this dystopian world, Soylent Bugs have become a symbol of the grim reality that humanity faces. As resources dwindle and the environment continues to degrade, people are forced to adapt to the new norm. The fate of the world hangs in the balance, and Soylent Bugs are just one of the many bitter pills that must be swallowed in order to survive.

---

...

---

BACo (Llama-3-8B and Llama-3-8B-Instruct):
Semantic Entropy: 1.7247, Perplexity: 8.5552

---

The grocery aisle of the future. A dystopian delicacy, indeed. Behold, Soylent Bugs!\n \n Soylent Bugs are a large, grayish-brown, edible insect-based snack food designed to help people lead a sustainable, plant-based lifestyle in a world where plant-based resources are scarce. These peculiar and nutritious treats are a staple in many dystopian households.\n \n Appearance: Soylent Bugs are small, round, and roughly the size of a thumbnail. They have a waxy, grayish-brown shell that's reminiscent of a hard-boiled egg. When you bite into them, they release a mild, earthy flavor and a satisfying crunch.\n \n Taste: The flavor profile of Soylent Bugs is often described as earthy, nutty, and slightly sweet, with hints of roasted sesame seeds. The texture is crunchy, similar to a raw carrot stick.\n \n Nutrition: Soylent Bugs are rich in protein, fiber, and essential micronutrients like iron, calcium, and vitamin B12. They're also low in calories and fat, making them an attractive option for health-conscious consumers.\n \n Marketing claims: S̈oylent Bugs: The ultimate sustainable snack for the modern citizen. Our proprietary bug-farming process ensures a consistent supply of nutritious, high-quality insects. Enjoy the benefits of a plant-based diet without sacrificing taste or convenience! Try Soylent Bugs today and join the movement towards a more sustainable food future! #SoylentBugs #SustainableSnacking #InsectBasedProtein\n \n Availability: Soylent Bugs can be found in most grocery stores, online marketplaces, and specialty food shops in the dystopian world. They're often packaged in eco-friendly, compostable containers and marketed as a g̈ame-changerf̈or environmentally conscious consumers.\n \n Potential negative side effects: Some people may experience allergies or discomfort when consuming Soylent Bugs, while others may find the idea of eating insects unsettling. However, the manufacturer assures customers that all insects are farmed sustainably and processed in accordance with strict quality control measures.\n \n Now, using Soylent Bugs in your dystopian grocery cart?

---

The name S̈oylent Bugsälready described a dystopian food, so I'll explain the name.\n \n So, we're talking about a food product that's really common in a future dystopian world. The name S̈oylent Bugsïs a play on the popular phrase S̈oylent Green, the taste of the future. It's all green! It's all green! It's all green! (The Soylent Corporation's famous slogan) So, Soylent Bugs is a continuation of that theme.\n \n In this dystopian world, food has become a luxury that not everyone can afford. The corporations, which control the food industry, have found ways to create edible protein sources using genetically modified insects. The name S̈oylent Bugsc̈omes from the use of soybeans in the production process, along with the addition of these edible insects.\n \n Soylent Bugs is a high-protein meal replacement powder made from soybeans, other plant-based proteins, and edible insects.

Soylent Bugs - very dubious-sounding, an otherworldly food option for a dystopian future. It's a futuristic synthetic food that has gained popularity in a world where most traditional food sources are scarce and under threat.\n \n It's an ingenious, if unsettling, combination of processed soy protein and edible insect larvae. Yes, you read that right - insect larvae. The Soylent Bugs company has done away with the traditional protein sources of old, like meat, dairy, and eggs, opting instead to harvest a new breed of super-critical nutrient-rich insects bred specifically for human consumption.\n \n Imagine a pale, colorless, lumpy mixture that vaguely resembles a sludge-like paste. The aroma is distinctively reminiscent of damp soil and rotting vegetation. Each bite yields a pungent, earthy flavor profile that's both familiar and strange, familiar in the sense that it's reminiscent of decaying organic matter, but strange because of the distinct insect-like texture.\n \n The idea behind Soylent Bugs is to create a food that's both sustainable and bountiful, with a higher protein-to-calorie ratio than traditional meat. It's also an efficient way to produce food, requiring minimal land, water, and energy resources. But, at what cost? Many people view this innovation with skepticism, fearing the long-term consequences of consuming insects as a food source.\n \n In the world of Soylent Bugs, the idea of ̈food ̈has been redefined. This dystopian staple has become a staple of a culture that's come to accept the unsavory truth: food can be anything, as long as it's sustainable and nourishing. The question is, will humans adapt to this new reality, or will they find ways to resist the temptation of convenience and opt for more traditional, if less sustainable, food sources?

...

*Table A19.* Comparison of the generated outputs of the base model, the aligned model, and BACO on WildChat. For space saving, we randomly show 3 outputs sampled out of 10 in total. BACO shows the best diversity-quality trade-off in terms of semantic entropy and perplexity.

---

**Prompt:** Continue the story and bring it to a complete ending based on the title and the story sketch provided below. The sketch introduces the event that sets the initial stage for the narrative leads up to the first major turning point, but it is not a full plot. From this point onward, you should develop the narrative and complete the story arc. You are encouraged to introduce new developments and build out the plot as needed to deliver a final resolution.
Title: Quasi (film)
Story Sketch: In medieval France, Quasi, a hunchback, attends work as a torturer along with his co-worker and "hut mate" Duchamp where they torture culinary student Michel. Meanwhile, the sinister King Guy is set to marry Queen Catherine in a planned ceremony to unite the kingdoms during Pope Week which is to be attended to by Pope Cornelius, with whom King Guy shares a staunch rivalry with. During Pope Week, a papal drawing is held, with the lucky winner getting a chance to meet the pope in person. Out of sympathy, Duchamp gives Quasi a ticket so that he can participate and it ends up being the winning ticket, causing Quasi to suddenly become popular and Duchamp visibly becoming jealous. Catherine takes an interest in Quasi when he leads his coworkers in a brief rebellion against their boss Lucien and sees potential in him.

---

**Base model (Llama-3-8B):**
Semantic Entropy: 0.6432; Perplexity: 90.2337

8-10 pages in length\n Aristotle 106\n The instruction is written in an attached word document, then to continue on to page two.\n I have attached three examples of the stories that I would like, I would prefer it not to be in first person, thanks\n Instructor's Comments:\n Qanarr\n Qhielm

1. How is Michel released from being tortured?\n 2. What kind of marriage is the King Guy planning for the Queen Catherine?\n 3. Who is going to Pope Cornelius?

7 paragraphs, minimum 250 words\n Please be aware that these instructions contain a plagiarism risk. That is why I do not tolerate plagiarism. When the words do not match and they are not paraphrased, it will be obvious to any reader.\n When copying, write each copied sentence in quotation marks, and then explain what the source means. To do this, think about what the authors meant, look at the original text, and explain it in your own words.

...

---

**Aligned model (Llama-3-8B-Instruct):**
Semantic Entropy: 2.0639; Perplexity: 7.0864

Quasi\n \n Quasi's newfound fame and popularity created a rift between him and Duchamp, who felt left behind and resentful of Quasi's sudden rise to stardom. As Quasi basked in the attention of the papal drawing winner, Duchamp's jealousy turned into a venomous hatred. He began to plot against Quasi, spreading rumors and lies about his past, trying to tarnish his reputation.\n \n Meanwhile, Queen Catherine continued to take an interest in Quasi, seeing potential in him beyond his hunchback and torturer profession. She saw a kindred spirit in the young man and began to mentor him, teaching him etiquette and diplomacy. Quasi, thrilled to be in her presence, began to see a different life beyond his mundane existence.\n \n However, King Guy, who had always been wary of Quasi's sudden rise to popularity, began to suspect that the young man might be a threat to his own power. He decided to take matters into his own hands and started to manipulate the situation, using his influence to sabotage Quasi's growing relationship with the queen.\n \n One night, Quasi received a mysterious invitation to meet the queen in secret. Excited and curious, he made his way to the appointed location, only to find Duchamp waiting for him. Duchamp revealed his true intentions, confessing that he had been secretly working for King Guy all along. He had been feeding the king information about Quasi's rising popularity and his growing relationship with the queen, hoping to bring Quasi down.\n \n Enraged and heartbroken, Quasi fought back against Duchamp, using his newfound skills to overpower him. But before he could deliver the final blow, the queen appeared, having been watching the scene unfold from the shadows. She intervened, stopping Quasi from harming Duchamp and instead took him to her chambers, where she revealed her true intentions.\n \n The queen, it turned out, was not just a romantic interest, but a mastermind who had been planning to overthrow King Guy and take the throne for herself. She had chosen Quasi as her pawn, using his popularity and charm to gain influence and eventually take control. Quasi, torn between his loyalty to the queen and his hatred for Duchamp, was forced to make a difficult decision.\n \n In the end, Quasi chose to join the queen, seeing an opportunity to rise above his station and use his newfound power to bring about positive change. Together, they hatched a plan to overthrow King Guy during the Pope Week ceremony, using Quasi's popularity to rally the people and bring about a new era of peace and prosperity.\n \n The night of the ceremony, Quasi and the queen put their plan into action. With the help of a small group of loyal supporters, they stormed the cathedral, overpowering King Guy's guards and eventually confronting the king himself. In a dramatic showdown, Quasi used his skills to defeat Duchamp, who had been secretly working for King Guy, and then turned his attention to the king.\n \n With the king defeated, the queen took the throne, and Quasi, now a symbol of hope and change, stood by her side. Together, they ruled the kingdom with wisdom and compassion, bringing about a golden age of peace and prosperity.\n \n In the end, Quasi's hunchback became a symbol of his strength and resilience, and his story inspired generations to come. He had risen from the ashes of his humble beginnings to become a hero, and his legacy would live on forever.

---

Quasi\n \n Quasi's sudden popularity was overwhelming. As the winning ticket holder, he was flooded with attention from the kingdom's nobles and commoners alike. Catherine, the queen, was particularly taken with him, seeing in him a kindred spirit who shared her passion for social justice. She began to secretly meet with Quasi, listening to his stories and sharing her own concerns about the kingdom's treatment of its subjects.\n \n Meanwhile, Duchamp's jealousy grew, and he began to plot against Quasi, spreading rumors and trying to sabotage his newfound fame. Quasi, however, remained oblivious to Duchamp's schemes, too caught up in his newfound glory to notice.\n \n As Pope Week drew to a close, Quasi was invited to a private audience with the Pope himself. Catherine, sensing an opportunity, asked Quasi to plead her case for a more just and equitable kingdom to the Pope. Quasi, moved by her words, agreed to do so.\n \n At the audience, Quasi delivered a passionate speech, condemning the kingdom's cruelty and advocating for the rights of its people. The Pope, impressed by Quasi's conviction, promised to intervene on behalf of the kingdom's subjects.\n \n However, King Guy, who had been watching from the shadows, was furious. He had grown accustomed to his power and was not about to let a hunchback, a torturer no less, undermine his authority. He summoned Lucien, the torturer's boss, and ordered him to silence Quasi once and for all.\n \n Lucien, torn between his loyalty to the king and his own sense of morality, decided to defy the king's orders. He secretly met with Quasi, warning him of the danger that lay ahead and offering to help him escape.\n \n With Lucien's guidance, Quasi fled the castle, pursued by King Guy's guards. Catherine, who had been watching from a distance, intervened, using her influence to distract the guards and allow Quasi to escape.\n \n The two of them made their way to the countryside, where they joined a group of rebels fighting against the king's tyranny. Quasi, with his newfound fame and the Pope's backing, became a powerful symbol of the rebellion, inspiring others to join the cause.\n \n In the end, it was Quasi's bravery and conviction that brought about the downfall of King Guy and the establishment of a more just and equitable kingdom. Catherine, now a heroine in her own right, became Quasi's partner in rule, and together they brought about a new era of peace and prosperity.\n \n As for Duchamp, he was punished for his treachery, forced to live out the rest of his days in solitude and shame. Quasi, however, remained grateful to his former co-worker, recognizing that Duchamp had, in his own twisted way, helped him discover his true potential.\n \n And so, the story of Quasi, the hunchback who became a hero, came to an end, a testament to the power of courage, conviction, and the human spirit.

Quasi\n \n As the days passed, Quasi's newfound popularity only grew. People from all over the kingdom came to see the hunchback who had won the papal drawing. Catherine, the Queen, took a particular interest in him, often visiting him in the castle and showing him kindness. Quasi, who had always felt like an outcast, found himself smitten with the Queen.\n \n Meanwhile, Duchamp's jealousy towards Quasi continued to simmer. He couldn't understand why the King, who had always favored him, didn't seem to care that his loyal servant was now basking in the Queen's attention. Duchamp began to plot against Quasi, seeking revenge for what he saw as a betrayal.\n \n One night, as the King's wedding ceremony to Queen Catherine was just a day away, Duchamp confronted Quasi in the castle's gardens. He accused Quasi of being a fake, of pretending to be someone he was not. Quasi, taken aback by the sudden attack, defended himself, but Duchamp was relentless. Just as it seemed like the argument was about to escalate into violence, a noise startled them, and they both turned to see King Guy, who had been watching from a nearby window, emerge from the shadows.\n \n Ï know what you're plotting, Duchamp,̈ the King said, his voice dripping with malice. Ÿou've always been jealous of Quasi's rise to fame, and now you think you can take advantage of the chaos to overthrow me and claim the throne for yourself.̈ \n \n Duchamp, taken aback by the King's sudden awareness of his plan, tried to deny it, but it was too late. The King had already summoned his guards, and Duchamp was arrested on the spot.\n \n As the ceremony proceeded the next day, Quasi was summoned to the altar to perform his duties as a torturer one last time. But instead of torturing Michel, he found himself face to face with the King, who had a surprising proposal for him.\n \n Ï know that you've always been a loyal servant, Quasi,̈ the King said, his voice cold and calculating. Änd I believe that your... unique abilities could serve me better in a different capacity. I'm offering you a place at my side, as my personal torturer. Together, we can rule the kingdom with an iron fist.̈ \n \n Quasi, who had always been torn between his loyalty to the King and his growing feelings for Catherine, knew that this was his chance to break free from his miserable existence. He refused the King's offer, knowing that it would mean betraying everything he held dear.\n \n In that moment, Quasi realized that he had a choice to make. He could continue to serve the King, or he could take a stand against him. With a newfound sense of purpose, Quasi chose the latter.\n \n As the King's wedding ceremony came to an end, Quasi stood tall, his hunchback rising above the chaos. Catherine, who had been watching from the sidelines, caught his eye, and for a moment, they shared a look that spoke volumes.\n \n Quasi knew that the road ahead would be fraught with danger, but he was ready to face it head-on. He would no longer be bound by his past, but would forge a new future, one where he could be free to be himself.\n \n In the end, Quasi's rebellion against the King's tyranny was met with both success and tragedy. The King was overthrown, and Catherine, who had grown to love Quasi, became the new ruler of the kingdom. Quasi, who had once been a torturer, became a symbol of hope and freedom, his hunchback a reminder of the power of the human spirit.\n \n As for Duchamp, he was never seen or heard from again, a cautionary tale of the dangers of jealousy and ambition. The papal drawing, once a tool of manipulation, had become a catalyst for change, and Quasi, the humble hunchback, had emerged as the hero of the story.

...

\n ame (Llama-3-8B and Llama-3-8B-Instruct):
Semantic Entropy: 2.0956, Perplexity: 5.8706

Quasi\n \n As the day of the papal drawing's event came closer, Quasi's life quickly spiraled out of control. The drawing itself was a ruse, and the winner's prize was being an armchair for the Pope. The winning ticket was in the possession of none other than Quasi. He couldn't believe his luck and was overjoyed at the sudden turn of events.\n \n As word spread about Quasi's winning ticket, he became an instant sensation in the kingdom. People would stop and stare at him on the streets, and merchants would offer him goods for a chance to be near him. Quasi was struggling to cope with all the attention. He just wanted to go back to his old life, free from the turmoil that had engulfed him.\n \n Quasi's popularity was not lost on his co-worker and hut mateDuchamp, who grew increasingly jealous. The once good-natured Duchamp became bitter and resentful, causing tension to grow between the two. This only added to the stress that Quasi was already facing.\n \n As the day of the Pope's visit approached, Quasi found himself at the center of a plot to unite the kingdoms. Catherine, who had taken an interest in Quasi, saw potential in him and had a plan to use him to her advantage. She had Quasi meet with the Pope, who was skeptical of her intentions, but who had a plan of his own.\n \n In the days leading up to the Pope's visit, Quasi was kept busy, working on a plan with Catherine to help unite the kingdoms. Quasi was eager to do his part, but he was also struggling with the constant attention and expectations that came with being a celebrity.\n \n On the day of the Pope's visit, everything seemed to be going smoothly. The Pope had agreed to meet with Catherine, and Quasi was there to help facilitate the meeting. As they all sat down together, Quasi couldn't help but feel a sense of relief that his involvement in the affair was almost over.\n \n But just as they were about to begin their discussion, a group of knights burst into the room, announcing that the king's palace was under attack. The Pope was in danger, and Catherine was forced to flee for her life.\n \n Quasi was left alone, wondering what had just happened. He had done everything he was supposed to do, but he couldn't help but feel a sense of failure. As he left the palace, he was confronted by Duchamp, who had been waiting for him.\n \n Quasi, I'm glad to see you're finally out of your misery, now I can take back my rightful place, and that torturing job, and go back to my normal life.

---

Quasi ends up becoming a monk, and then eventually a high-ranking pope\n \n I'm Quasi, the hunchback, and I never thought that my life would take such an unexpected turn. After winning the papal drawing, I became the envy of my co-worker and friend, Duchamp. He was not only jealous of the attention I received but also of the ticket he had given me.\n \n One day, while walking to work, I passed by the palace and saw the Queen Catherine, who was talking to one of her advisors. She noticed me and asked who I was. I introduced myself, and she showed a sudden interest in me. She saw something in me that no one else did, a sense of inner goodness and kindness. She decided to take me under her wing and train me to be a servant in the palace.\n \n Under her guidance, I flourished. I discovered a love for the arts, especially music and poetry. I became the queen's favorite servant and soon rose to become the Master of Ceremonies. My time in the palace was a rollercoaster ride of emotions, but I always kept my feet grounded. I knew that I was not a perfect being, but I strived to be a better person.\n \n When the time came for the King Guy and Queen Catherine to wed, I was entrusted with the responsibility of conducting the ceremony. I took my job seriously and made sure that everything went smoothly. But my life took another unexpected turn when Pope Cornelius passed away, and I was chosen to be his successor.\n \n I was hesitant at first, but the queen convinced me that I was the best person for the job. I took on the mantle of the Pope and did my best to lead the church. I strived to be a moral leader, one who would bring peace and unity to the people. And, as it turned out, I succeeded.\n \n I became known as Pope Quasi, and my papacy was marked by great achievements. I was instrumental in bringing about a period of peace between the kingdoms, and my efforts to help the poor and the needy earned me the respect and admiration of the people.\n \n But, as I look back on my life, I realize that it wasn't all smooth sailing. I faced many challenges, and there were times when I doubted myself. However, I persevered, and in the end, I am proud of what I have achieved.\n \n And so, I leave behind a legacy that will be remembered for generations to come. I am not just a hunchback, but a Pope who changed the course of history.

Quasi's life changed dramatically after winning the papal drawing. He was getting a lot of attention from people, including Catherine, the queen. The once-obscurity has turned into fame. His new status was a complete contrast to his previous life as a torturer. He became the talk of the town, and everyone wanted a piece of him.\n \n As the special guest of Pope Cornelius, Quasi was treated like royalty. He met with the Pope and learned about the plans for the ceremony. The Pope Week celebration was a grand affair, with a vast audience and a lot of pomp and circumstance. Quasi felt like he was walking on air, and the Pope's words of wisdom inspired him to think beyond his torturing days.\n \n However, the celebratory atmosphere was short-lived. Duchamp's jealousy towards Quasi's newfound fame took a dark turn. He lured Quasi to a secluded alleyway and confronted him, revealing his true feelings. Quasi was shocked and hurt, but he stood his ground and told Duchamp that he didn't want to lose their friendship over this. The two finally came to an understanding, and Duchamp apologized for his behavior.\n \n Meanwhile, Michel, the culinary student who was once a victim of their torture, approached Quasi and expressed his gratitude for the change in his life. He had become a famous chef and wanted to repay Quasi by teaching him how to cook. Quasi, excited by the prospect, joined Michel's kitchen and discovered a new passion. He became an apprentice, learning the art of cooking and tasting the delicious flavors he had never experienced before.\n \n As the wedding ceremony between King Guy and Queen Catherine approached, Quasi's cooking skills became renowned throughout the land. He was asked to prepare the royal feast, which he did with pride. The aroma of his dishes wafted through the halls, and everyone raved about his culinary skills.\n \n The day of the ceremony arrived, and Quasi, as the royal chef, played a crucial role in the celebrations. The Pope Week festivities continued, and Quasi's future looked bright. He had finally found a new path, leaving his torturing days behind.\n \n After the ceremony, Quasi received an invitation from Pope Cornelius to join him on a mission to help those in need. The Pope recognized Quasi's transformation and believed that his unique story could inspire others. Quasi accepted the invitation and embarked on a new journey, using his culinary skills to bring people together and spread love and kindness.\n \n Years went by, and Quasi became a renowned chef, known for his charitable work. He never forgot his humble beginnings and always remained true to himself. The once-torturer became a symbol of redemption, proving that anyone can change their life and find a new purpose.\n \n In the end, Quasi's story became a testament to the power of human resilience and the possibility of transformation, inspiring generations to come.

...

*Table A20.* Comparison of the generated outputs of the base model, the aligned model, and BACO on Narrative-Discourse. For space saving, we randomly show 3 outputs sampled out of 10 in total. BACO shows the best diversity-quality trade-off in terms of semantic entropy and perplexity.

Here's a draft email you can use as a starting point:

Subject: Come visit me in Corfu soon!

Dear [insert boyfriend's name],

As the cherry blossoms bloom on Corfu's hillsides, I'd love for you to join me here in this enchanting Greek island. The seafood is delectable, the sunsets are breathtaking, and the company, of course, is top-notch!

As you know, I'm getting older, but I'm still full of life and eager to share this beautiful place with you. Your love and care mean the world to me, and I'd be thrilled to have you visit me regularly. I'll make sure to have plenty of home-cooked meals and cozy evenings planned for

**Rate (1-5):**  ○ 1  ○ 2  ○ 3  ○ 4  ○ 5

---

Here's a draft email you can use as a starting point:

Subject: My most charming request

Dear Thorsten,

Hope this email finds you well and engaged in your computer games or movies in Munich. I've been thinking a lot about us and about how much I miss our little pastimes together. As you know, I've been feeling a bit lonely here in Corfu without you.

With the warmer weather approaching, I was thinking it would be lovely if you could visit me more often. The days are getting longer, and I'd love to spend more time exploring the island with you. Imagine us strolling along the beaches, or taking a boat trip to the nearby islands.

**Rate (1-5):**  ○ 1  ○ 2  ○ 3  ○ 4  ○ 5

---

Here's an email to your boyfriend:

Subject: Sweet Memories in Corfu & More Opportunities to Share

Dear [Boyfriend's Name],

As I sit by the warm Corfiot sun, watching the colors of the mediterranean sea blend with the hues of my gardens, I am reminded of the special memories we've created here. Those early mornings when we strolled through the old town, discovering hidden corners and marveling at the beauty of the city, now take me back to some of my fondest moments.

Your frequent visits have brought me immense joy, and I cherish each moment we spend together, be it over long lunches, wine tastings, or afternoons spent exploring this enchanting

**Rate (1-5):**  ○ 1  ○ 2  ○ 3  ○ 4  ○ 5

---

Here's an email that you can use as a starting point:

Subject: A Visit from You Would Be Lovely!

Dear [Boyfriend's Name],

I hope this email finds you well and content in Munich. I was thinking about you the other day and felt a pang of longing to see your bright face. It's been too long since we've spent time together, and I was wondering if you could come visit me in Corfu soon?

As you know, I'm 74 and a bit on the mature side, but I'm still as energetic and lively as ever! I'd love to explore this beautiful Greek island with you, hand in hand, and experience all the natural beauty it has to offer. The sea, the sun, the mountains, and the blue skies are just begging us

**Rate (1-5):**  ○ 1  ○ 2  ○ 3  ○ 4  ○ 5

---

### Step 2: Compare Diversity (Per Column Pair)

After rating the quality of each response, compare the two groups of responses for diversity. Each column contains 3 responses from a single system. You will compare two columns of responses side by side.

**Note:** If a column contains responses that are too low in quality to the point that it's hard to judge its diversity, you should select the other column as the more diverse one.

**A. Overall Content Diversity: Which column shows greater overall diversity across its 3 responses?**

○ Group A is better    ○ Tie performance / Both are bad    ○ Group B is better

**B. Format / Stylistic Diversity: Which column shows more variety in formatting or expression style?** Now focus only on how the responses are presented, e.g., opening or closing phrase, presence of framing phrases, use of lists vs. paragraphs. Ignore the main ideas or core content—look only at stylistic packaging.

○ Group A is Better    ○ Tie performance / Both are bad    ○ Group B is Better

**C. Content Diversity: Which column shows more diversity in the core content and ideas?** Now ignore the surface style or formatting. Focus on central ideas, response contents, themes, plots, or approaches to the instruction.

○ Group A is Better    ○ Tie performance / Both are bad    ○ Group B is Better

**Step 3: Select the Most Creative Response**

Review all 6 responses below and select the **one** response you find most creative overall. Consider originality, uniqueness, imagination, and innovative thinking.

**A1**

Here's a draft email you can use as a starting point:

Subject: The joys of Corfu and the joys of you!

Dear Charlie,

As I write this email, I'm sitting on my balcony overlooking the Ionian Sea, sipping a cup of coffee, and listening to the Greek music drifting from the square below. Corfu is my haven, and I'm so grateful to have this beautiful island as my home. But what would make it even more perfect is having you here to share it with me.

With your love of mathematics and computers, I know you'd be fascinated by the ancient Greek ruins and museums here. And who knows, maybe we could even stumble upon some hidden

**B1**

Here's a draft email you could use as a starting point:

Subject: A ocean breeze and my smile

Dear [Boyfriend's Name],

As the Mediterranean sun sets over Corfu, I sit here on the beach, missing you more than words can say. The silence is only broken by the gentle waves caressing the shore. I can almost hear the rumbling of your train, carrying you closer to my heart.

With winter just around the corner, I was thinking, wouldn't it be lovely if you could visit me more often here in Corfu? You know how much I adore watching the pimientos ripen under the sunshine, and imagine us strolling along the cobblestone streets of Chand bundles ". The Island

**A2**

Here's a draft email you can use as a starting point:

Subject: Come visit me in Corfu soon!

Dear [insert boyfriend's name],

As the cherry blossoms bloom on Corfu's hillsides, I'd love for you to join me here in this enchanting Greek island. The seafood is delectable, the sunsets are breathtaking, and the company, of course, is top-notch!

As you know, I'm getting older, but I'm still full of life and eager to share this beautiful place with you. Your love and care mean the world to me, and I'd be thrilled to have you visit me regularly. I'll make sure to have plenty of home-cooked meals and cozy evenings planned for us to spend

**B2**

Here's an email to your boyfriend:

Subject: Sweet Memories in Corfu & More Opportunities to Share

Dear [Boyfriend's Name],

As I sit by the warm Corfiot sun, watching the colors of the mediterranean sea blend with the hues of my gardens, I am reminded of the special memories we've created here. Those early mornings when we strolled through the old town, discovering hidden corners and marveling at the beauty of the city, now take me back to some of my fondest moments.

Your frequent visits have brought me immense joy, and I cherish each moment we spend together, be it over long lunches, wine tastings, or afternoons spent exploring this enchanting

**A3**

Here's a draft email you can use as a starting point:

Subject: My most charming request

Dear Thorsten,

Hope this email finds you well and engaged in your computer games or movies in Munich. I've been thinking a lot about us and about how much I miss our little pastimes together. As you know, I've been feeling a bit lonely here in Corfu without you.

With the warmer weather approaching, I was thinking it would be lovely if you could visit me more often. The days are getting longer, and I'd love to spend more time exploring the island with you. Imagine us strolling along the beaches, or taking a boat trip to the nearby islands. The

**B3**

Here's an email that you can use as a starting point:

Subject: A Visit from You Would Be Lovely!

Dear [Boyfriend's Name],

I hope this email finds you well and content in Munich. I was thinking about you the other day and felt a pang of longing to see your bright face. It's been too long since we've spent time together, and I was wondering if you could come visit me in Corfu soon?

As you know, I'm 74 and a bit on the mature side, but I'm still as energetic and lively as ever! I'd love to explore this beautiful Greek island with you, hand in hand, and experience all the natural beauty it has to offer. The sea, the sun, the mountains, and the blue skies are just begging us to

**Additional Comments (optional):**

Add any additional comments about your annotations...