# OpenReview forum: "Optimizing Diversity and Quality through Base-Aligned Model Collaboration"
_ICML.cc/2026/Conference — ICML 2026 regular_

### Official Review · Reviewer_mnra · 2026-02-18

**Soundness:** 4
**Presentation:** 3
**Significance:** 3
**Originality:** 3
**Overall Recommendation:** 5
**Confidence:** 2

**Summary:**

This paper introduces a new method to balance diversity and quality in LLMs generation. It is based on the observation that (i) Base LLMs are often strong in diversity but weak in quality, and (ii) Aligned (e.g., through Instruction-tuning) LLMs are strong in quality but weak in diversity. It designs a decoding method that makes the two models collaborate through a router that allows switching the choice of the two models for sampling. Several designs are experimented with for the router choice, which is non-trainable. Performance on the Diversity / Quality space is then measured both in the lexical and semantic space, using both coverage and dominance as metrics, demonstrating the competitiveness of the method.

**Compliance With Llm Reviewing Policy:**

Affirmed.

**Final Justification:**

The authors did a great job during the rebuttal. I believe the paper is of high quality, and I maintain my initial recommandation (accept).

**Key Questions For Authors:**

See the questions (Q1, Q2, and Q3) above.

I find the scores for the row decoding in Table 2 surprisingly low. Q4: How did you choose the beam size and the diversity penalty for the Diverse Beam Search baseline? For the base/aligned vanilla sampling, can you specify which truncation method did you use (top-k or top-p) ?

**Limitations:**

The authors acknowledge in the text that the improvement in the text is great in the semantic front, although sometimes weaker in the lexical part (e.g., looking at the lexical dominance in Tables 2 and 4). I think the manual design of the Router is also a limitation. It would be valuable to add a full limitations section describing all the weaknesses of the method at the end.

**Strengths And Weaknesses:**

*Soundness*

The paper is well written and addresses the important problem of balancing quality and diversity in LLMs generation.

The submission is sound, and the paper is rigorous. The experimental setup is complete with numerous experimental details described in the Appendix.

*Presentation*

Use $t-1$ in L.134.

Can you improve the resolution of the Figures in the paper?

*Significance*

The results are promising, showing in particular substantial improment on the Semantic axis. However, I have a question (Q1) regarding the assertion "while it fails on semantic metrics such as Semantic Entropy": -> but the coverage and dominance of -Rand is still competitive with respect to -P and -FC in the results of Table 3, right? Q2: How do you explain such bad scores for -P and -FC compared to the -Rand variant (Table 3) ? Q3: How did you choose $\\gamma$ ?

---

> ### Author Rebuttal · Authors · 2026-03-31
>
> We greatly appreciate the reviewers' recognition of these strengths: our paper addresses an "important" topic (hL6H, mnra). Our method is "well motivated" (hL6H, UTn3) and "simple and effective" (RWdq). The paper is "well written" (hL6H, UTn3, mnra) and "sound and rigorous" (mnra). The experiments and evaluation are "comprehensive" (hL6H, RWdq) and "complete" (mnra), demonstrating "promising" results with "substantial improvement" (mnra).
>
> ---
>
> **Q1 and Q2: -RAND Performance**
>
> Thank you for this observation. From the data, -RAND does show some lead over -P/-FC in Dominance. However, we note that Dominance is a relative indicator computed against the global Pareto frontier of all methods. Since combinational routers (-P-PUNC, -H-PUNC, -P-FC) dominate much of this frontier, they absorb Dominance share that would otherwise be attributed to single-strategy routers. In a pairwise comparison between -RAND and -P/-FC alone, the single-strategy routers would show more significant Dominance. We acknowledge that -RAND achieves comparable Coverage to -P/-FC. We attribute this to the fact that single-strategy routers (-P, -FC) rely on a single signal and can be aggressive in their routing, which limits their aggregate coverage. That said, when compared against the stronger combinational routers (e.g., -P-PUNC), -RAND's limitations become clear.
>
> Overall, we stress that **diversity is subjective and multidimensional**. Different people weigh metrics differently (notably, Reviewer UTn3 emphasizes lexical diversity while Reviewer RWdq considers Distinct-1 "less valuable" than Semantic Entropy). Rather than making a subjective choice, we chose to include as many metrics as possible from the literature and optimize toward the joint Pareto frontier. A method that wins only on lexical diversity is unlikely to reflect a meaningful diversity improvement in most practical contexts, since random or nonsensical text can achieve high lexical variation without introducing substantive semantic differences. -RAND is under this case, whose gains are confined to lexical metrics (e.g., Dist-1), achieving only 0% Dominance on Semantic Entropy. This leads to the view that unguided switching injects surface-level randomness without inducing semantically-meaningful diversity (see §5.2 Line 366 (R) and Appendix J for discussion and examples). -P-PUNC, by contrast, achieves clear diversity dominance on both lexical and semantic metrics.
>
> Moreover, we would like to clarify that **our framework is the contribution, not any single router or exact set of metrics**. We demonstrate that inference-time token-level model collaboration between a base and an aligned model is effective for optimizing the diversity-quality trade-off, rather than advocating for specific routers or metrics. The family of routers we present all outperform other baselines on the Pareto frontier of the set of metrics we explored, indicating that the BACo framework is easy to apply without heavy tuning of the routers.
>
> ---
>
> **Q3: $\gamma$ Selection and Meaning**
>
> We do not use a fixed $\gamma$ in our experiments. Instead, BACo is evaluated across a range of $\gamma$ values (e.g., [0.1, 0.2, ..., 0.9] for -P), and our evaluation comprehensively assesses the entire diversity-quality trade-off along different spectrums. Coverage and Dominance (§4.1, Figure 1B) are designed to aggregate performance over all $\gamma$ values: Coverage measures the AUC of the trade-off curve, and Dominance captures the portion of the global Pareto frontier attributed to each method. This ensures that our results reflect general improvement across the full range, not a single tuned operating point.
> In practice, $\gamma$ function, analogously to temperature, can be a user-facing hyperparameter: larger $\gamma$ biases toward the base model (more diversity) and smaller $\gamma$ biases toward the aligned model (more quality), as illustrated in Table A19. A method with strong performance across the entire frontier can be broadly applied to different tasks or user intents.
>
> ---
>
> **Q4: Baseline Hyperparameters**
>
> For experiments other than the baseline of increasing temperature, we by default set temperature = 1.0 and top-p = 0.9 without top-k, following the NoveltyBench [1]. Diverse Beam Search hyperparameters are beams = 2n = 10, beams-group = n = 10, diversity-penalty = 1.0, following the original paper [2], where n is the sample size of our experiments. We will make these parameters more explicit in Appendix H.
>
>
> **Limitations Section**
>
> Thank you for the suggestion. We will add a dedicated Limitations section consolidating all relevant discussions in the revision.
>
> [1] Zhang et al., "NoveltyBench: Evaluating language models for humanlike diversity." arXiv:2504.05228, 2025.
>
> [2] Vijayakumar et al., "Diverse beam search: Decoding diverse solutions from neural sequence models." AAAI 2018.

---

> > ### Author Rebuttal · Reviewer_mnra · 2026-04-02
> >
> > Thanks to the authors for their rebuttal. I believe the paper is quite solid and deserves acceptance.

---

> > > ### Author Response · Authors · 2026-04-03
> > >
> > > We thank Reviewer mnra for the acknowledgement and the positive assessment.

---

### Official Review · Reviewer_RWdq · 2026-02-27

**Soundness:** 3
**Presentation:** 2
**Significance:** 3
**Originality:** 3
**Overall Recommendation:** 4
**Confidence:** 4

**Summary:**

The paper studies the tension between diversity and quality in aligned LLMs and proposes BACo, an inference-time, token-level collaboration framework between a base model and its aligned counterpart. A lightweight router decides at each token which model to use, based on logit-based uncertainty (max probability, entropy) and content-based heuristics (punctuation, function words), with a tunable threshold controlling the diversity-quality balance. The authors evaluate BACo on three open-ended generation benchmarks and find that BACo dominates prior inference-time diversity methods and single-model baselines.

**Compliance With Llm Reviewing Policy:**

Affirmed.

**Final Justification:**

I maintain the positive score.

**Key Questions For Authors:**

1. Could you provide a precise algorithm for BACo decoding that reconciles Equation (1) with the actual sampling procedure?

2. Could you provide downstream results in math and code?

3. What is the inference overhead of BACo relative to an aligned-only baseline?

**Limitations:**

Yes

**Strengths And Weaknesses:**

Strengths

1. The paper targets a very concrete and currently under-served need: improving group-level output diversity of aligned LLMs without sacrificing quality and without retraining.

2. Simple and effective collaborative framework with controllability.

3. Comprehensive evaluation. The evaluation uses 11 diversity metrics and 2 quality metrics.
For each method, performance is visualized as a curve over control parameters (temperature, threshold), not a single point.


Weaknesses

1. The router is entirely heuristic, and the design space feels under-explored given the centrality of routing. Section 3.1 define a set of hand-crafted routers based on max probability thresholds, entropies, words, and punctuation. There is no principled justification or analysis of why these particular signals are optimal or robust, nor why the threshold parameterization is appropriate.  The paper does not explore any learned routers, despite referencing multi-objective optimization and RL-like literatures.

2. Evaluation risks over-aggregation and can obscure where BACo actually helps or hurts. Coverage averages indiscriminately across 11 × 2 spaces, even though some metric pairs are arguably redundant or less meaningful. For example, Semantic Entropy and NoveltyBench’s Distinct score are arguably more important than Distinct-1.

3. Mathematical formulation is fairly shallow and glosses over crucial implementation details of sampling.

---

> ### Author Rebuttal · Authors · 2026-03-31
>
> We thank all reviewers for recognizing BACo as a well-motivated, important, and effective method with comprehensive experiments.
>
> ---
>
> **W1: Characterization of routing as “heuristic”**
>
> First, we emphasize that the simplicity of our routing strategies is a deliberate strength of this work: they are easy to understand, offer direct control over behavior, and can be readily applied across different models and tasks without additional complexity.
>
> Second, such routers are well-established in prior work. Logit-based routing is used in Nudging [12], Co-LLM [13], and speculative decoding [14]; content-based routing is used in FUDGE [15] and Switch Generation [16]. Our designs are inspired by and informed by these works.
>
> Third, we clarify the motivation behind the two categories: (i) Logit-based routing (model uncertainty). Tokens with low maximum probabilities or high entropy indicate positions where the model is uncertain. These are natural candidate points for diversification [17, 18]. (ii) Content-based routing (linguistic features). Linguistic structures such as content words and punctuation serve as natural "branch points" in text. Triggering routing at these points produces meaningful, human-perceived diversity [19, 20].
>
> Combining both categories provides complementary coverage, which is why -P-PUNC achieves the strongest overall performance (§5.2).
>
> **Hyperparameter selection.** Each method is evaluated as a *curve* across a range of $\gamma$ values rather than a single operating point (§4.1, Figure 1B). Coverage is the AUC of this curve; Dominance is the portion of the global Pareto frontier attributed to each method. These indicators aggregate over all $\gamma$ values, so results reflect general outperformance, not a single tuned point. In practice, $\gamma$ function, analogous to temperature, is a user-facing hyperparameter: larger $\gamma$ biases toward the base model (more diversity) and smaller $\gamma$ biases toward the aligned model (more quality), as illustrated in Table A19.
>
> ---
>
> **W2: Metric Aggregation**
>
> **On the aggregated presentation.** The aggregation across metrics is to save space in the paper presentation. Raw metric scores for each method, hyperparameter, and dataset are available in the repository and will be linked after the anonymous period.
>
> **On whether some metrics are less meaningful.**
> Overall, we stress that diversity is **subjective and multidimensional**. Different people weigh metrics differently (notably, Reviewer UTn3 emphasizes lexical diversity while Reviewer RWdq considers Distinct-1 "less valuable" than Semantic Entropy). Rather than making a subjective choice, we include as many metrics as possible from the literature and optimize toward the joint Pareto frontier. We clarify that our framework is the contribution, not any single router or metric set. The family of routers we present outperforms baselines across the metrics explored in the Pareto frontier, indicating BACo is easy to apply without heavy tuning.
>
> ---
>
> **W3/Q1: Pseudocode for BACo Decoding**
>
> We will add a pseudocode box reconciling Eq.(1) with the actual sampling procedure in the revised manuscript.
>
> ---
>
> **Q2: Downstream Math and Code**
>
> §5.1 (Line 377 Left) and Appendix I include results on GSM8K (mathematical reasoning). And Figure A7 shows BACo achieves higher diversity at matched accuracy on GSM8K across multiple diversity metrics. Please refer to them for detailed results.
>
> Coding tasks are left for future work, as linguistic diversity metrics may not be appropriate indicators for code generation.
>
> ---
>
> **Q3: Inference Overhead**
>
> **Compute overhead comparison.** BACo requires two forward passes per decoding step. Unlike prompting baselines (in-context resampling, paraphrase prompting) that require n sequential decoding passes, diverse beam search (≥n beam expansions), or training-time methods, BACo's ~2× FLOPs overhead is a fixed multiplier independent of sample count. When running both models simultaneously, the wall-clock time approaches 1× despite 2× total FLOPs.
> Though our paper focuses on the idea and prototype, further engineering-level optimizations are complementary and applicable when deploying at scale, e.g., speculative decoding [1], LoRA-aligned models, and KV cache sharing (discussed in §A.4).
>
> **[New Experiment] Wall-clock runtime comparison.** We report wall-clock runtime under the same hardware (A100 80GB), focusing on inference-time methods for fair comparison. All methods use the HuggingFace library.
>
> ***Results:*** **Please refer to [Table 1](https://gist.github.com/anonymous-rebuttal-anonymous/4850f1fcea39eed8321034677a808daf?permalink_comment_id=6067929#gistcomment-6067929)**
>
> ***Findings:*** BACo's current prototype is slower than single-model baselines, but is comparable to or cheaper than some other baselines while significantly outperforming them. The implementation is not yet engineered for speed, and the optimizations above (§A.4) can further improve it.

---

> > ### Author Rebuttal · Reviewer_RWdq · 2026-04-01
> >
> > Thank you for the thoughtful rebuttal. While some aspects (e.g., principled routing design) could still be further explored, I think the current explanations sufficient for the scope of this paper. I therefore consider my concerns largely addressed and keep my positive score unchanged.

---

> > > ### Author Response · Authors · 2026-04-02
> > >
> > > We thank Reviewer RWdq for the acknowledgement and the positive assessment.

---

### Official Review · Reviewer_UTn3 · 2026-03-12

**Soundness:** 3
**Presentation:** 2
**Significance:** 2
**Originality:** 3
**Overall Recommendation:** 4
**Confidence:** 4

**Summary:**

The paper proposes an inference-time framework for base and aligned models to optimize both diversity and quality of responses during generation. It dynamically routes token generations between two models based on routing heuristics i.e, uncertainty-based and content-based rules. The paper evaluates BACO mainly on open-ended generation benchmarks and shows that BACO consistently achieves better results compared to baselines in terms of quality–diversity trade-offs.

**Compliance With Llm Reviewing Policy:**

Affirmed.

**Final Justification:**

Rebuttal addressed my concerns.

**Key Questions For Authors:**

1. How does the inference cost compared to the baselines in terms of runtime overhead?
2. Does switching between the base and aligned models during generation lead to noticeable changes in tone or coherence in the generated responses?
3. Could the authors analyze how much improvement is due to principled routing heuristics versus simply injecting randomness from the base model?

**Limitations:**

Yes

**Strengths And Weaknesses:**

Strengths:
1. The paper is well written and approach is easy to understand and analyze, and motivation for combining both base and aligned models is well explained.
2. The proposed approach operates at inference level and doesn't require costly retraining or modifying the model architecture.
3. The paper evaluates the method using both automated metrics and human evaluation, providing multiple perspectives on the diversity-quality trade-off.

Weaknesses:
1. Although, BACO is inference level approach, but it requires running two models during decoding and making routing decisions at each token step, which may introduce additional computational overhead. Authors didn't discussed that thoroughly.
2. While the alternate shifting between base and aligned model may improve diversity, it could potentially introduce inconsistencies in tone, or reasoning within the generated text. The paper does not analyze whether switching between models affects the coherence or consistency of the output.
3. According to Table 3 the best-performing routers are combinations like -P-PUNC, but the random router performs surprisingly well in some cases e.g., Lexical. That weakens the proposed approach a bit. If a random router is already competitive on these metrics, then the advantage of the proposed heuristics may be less principled than the paper suggests.

---

> ### Author Rebuttal · Authors · 2026-03-31
>
> We thank all reviewers for recognizing BACo as a well-motivated, important, and effective method with comprehensive experiments.
>
> ---
>
> **W1/Q1: Inference Cost**
>
> **Compute overhead comparison.** BACo requires two forward passes per decoding step. Unlike prompting baselines (in-context resampling, paraphrase prompting) that require n sequential decoding passes, diverse beam search (≥n beam expansions), or training-time methods, BACo's ~2× FLOPs overhead is a fixed multiplier independent of sample count. When the implementation runs both models simultaneously, the wall-clock time approaches 1× despite 2× total FLOPs.
> Though our paper focuses on the idea and prototype, further engineering-level optimizations are complementary and applicable when deploying at scale, e.g., speculative decoding [1], LoRA-aligned models, and KV cache sharing (discussed in §A.4).
>
> **[New Experiment] Wall-clock runtime comparison.** We report wall-clock runtime under the same hardware (A100 80GB), focusing on inference-time methods for fair comparison. All methods use the HuggingFace library.
>
> ***Results:*** **Please refer to [Table 1](https://gist.github.com/anonymous-rebuttal-anonymous/4850f1fcea39eed8321034677a808daf?permalink_comment_id=6067929#gistcomment-6067929).**
>
> ***Findings:*** BACo's current prototype is slower than single-model baselines, but is comparable to or cheaper than some other baselines while significantly outperforming them. The implementation is not yet engineered for speed, and the optimizations above (§A.4) can further improve it.
>
> ---
>
> **W2/Q2: Coherence and Tone Consistency**
>
> We provide evidence from complementary angles:
>
> **Automatic quality metrics.** Our existing quality metrics, e.g., perplexity, are widely used as a proxy for fluency [2,3,4,5,6], and reward scores from Skywork-Reward for holistic quality, including coherence [7]. BACo significantly outperforms baselines on the Pareto frontier on these metrics (§4.1) provides indirect evidence against quality degradation.
>
> **[New Experiment] Fluency, coherence, and consistency evaluation.** We conduct experiments aligned with §5.5 using UniEval [8] (fluency, coherence, consistency), BARTScore [9], and LLM-as-a-judge [10] (GPT-4o-mini, 1-5 scale). We compare: Aligned (temp=1.0) as the default baseline and quality ceiling, Aligned (temp=1.45), and BACo-P-Punc (threshold=0.2). The latter two are selected at match diversity levels for fair comparison.
>
> ***Results:*** **Please refer to [Table 2](https://gist.github.com/anonymous-rebuttal-anonymous/4850f1fcea39eed8321034677a808daf?permalink_comment_id=6067880#gistcomment-6067880).**
>
> ***Findings***: BACo maintains fluency, coherence, and consistency close to the aligned baseline at default temperature (e.g., UniEval-Fluency 0.847 vs. 0.840) while significantly improving diversity (Semantic Entropy 0.97→1.39). In contrast, increasing temperature causes severe degradation across all metrics (e.g., LLM-judge scores fall by 35-40%).
>
> **Human evaluation.** The human evaluation quality rubric (Table A12, Step 1) explicitly lists Fluency as an important quality criterion with a detailed description. Human evaluators give BACo a mean quality score of 4.04 vs. 2.83 for the aligned baseline (Table 5). No quality degradation was identified.
>
> **Qualitative evidence.** Appendix N includes examples.
>
> ---
>
> **W3/Q3: Random Router Performance**
>
> Overall, we stress that **diversity is subjective and multidimensional**. Different people weigh metrics differently (notably, Reviewer UTn3 emphasizes lexical diversity while Reviewer RWdq considers Distinct-1 "less valuable" than Semantic Entropy). Rather than making a subjective choice, we include as many metrics as possible from the literature and optimize toward the joint Pareto frontier. A method that wins only on lexical diversity is unlikely to reflect meaningful diversity improvement, since random or nonsensical text can achieve high lexical variation without substantive semantic differences. -RAND falls under this case: its gains are confined to lexical metrics (e.g., Dist-1), achieving only 0% Dominance on Semantic Entropy. This is consistent with that unguided switching injecting surface-level randomness without semantically meaningful diversity (§5.2 Line 366 (R), Appendix J). -P-PUNC, by contrast, achieves clear diversity dominance on both lexical and semantic metrics.
>
> **Our framework is the contribution, not any single router or metric set.** We demonstrate that inference-time token-level collaboration between a base and aligned model is effective for optimizing the diversity-quality trade-off, rather than advocating for specific routers or metrics. The family of routers we present all outperform baselines across the metrics explored, indicating BACo is easy to apply without heavy router tuning.
>
> **Citation**: [citations](https://gist.github.com/anonymous-rebuttal-anonymous/4850f1fcea39eed8321034677a808daf?permalink_comment_id=6067931#gistcomment-6067931).

---

> > ### Author Rebuttal · Reviewer_UTn3 · 2026-04-03
> >
> > Thank you authors for addressing my concerns. I'll raise my score.

---

> > > ### Author Response · Authors · 2026-04-03
> > >
> > > We thank Reviewer UTn3 for the acknowledgement and for raising the score. We are glad our response addressed your concerns.

---

### Official Review · Reviewer_hL6H · 2026-03-13

**Soundness:** 4
**Presentation:** 4
**Significance:** 2
**Originality:** 3
**Overall Recommendation:** 4
**Confidence:** 4

**Summary:**

The paper introduces BACO, an inference time decoding method that routes between the base model (e.g. Llama3-8b) and the aligned model (llama-3-8b-instruct) to achieve better quality-diversity tradeoffs. Experiments on wildchat & noveltybench show that the propose method is effective.

**Compliance With Llm Reviewing Policy:**

Affirmed.

**Key Questions For Authors:**

Could you share the amount of resources needed for running the method?

Could you share whether this works on multi-turn settings, could be as simple as mtbench.

**Limitations:**

Yes.

**Strengths And Weaknesses:**

Strengths:
- The paper is well written, very easy to understand and well motivated.
- For a decoding method, the experiments were comprehensive with many detailed ablations. The evaluation is also very comprehensive.
- The topic that the paper studies is important. It is a training-free method to improve the pareto front between quality and diversity. Could be useful for many downstream applications.

Weaknesses:
- **Scalability**: So co-decoding from 2 LMs requires you to maintain 2 KV Caches and serve 2 models. The overhead of this method becomes significant. Essentially this is 2x the cost of decoding from a single LM. This make me question whether people are going to adopt this especially when the LMs are large and / or the generations becomes super long.

- The paper only considers single turn chatting tasks, whereas the majority usage is multi-turn agentic tasks. It is unclear to me whether the method would still work under a different setting. Also a good use of diverse responses is to do RL, but with the cost so high and the method significantly off-policy, I doubt that it would be useful in RL.

---

> ### Author Rebuttal · Authors · 2026-03-31
>
> We greatly appreciate the reviewers' recognition of these strengths: our paper addresses an "important" topic (hL6H, mnra) that "targets a very concrete and currently under-served need" (RWdq). Our method is "well motivated" (hL6H, UTn3), "simple and effective" (RWdq), and "useful for many downstream applications" (hL6H). The paper is "well written" (hL6H, UTn3, mnra), "very easy to understand" (hL6H, UTn3), and "sound and rigorous" (mnra). The experiments and evaluation are "comprehensive" (hL6H, RWdq) and "complete" (mnra), "providing multiple perspectives" (UTn3) and demonstrating "promising" results with "substantial improvement" (mnra).
>
> ---
>
> **W1/Q1: Resource Requirements**
>
>
> **Compute overhead comparison.** BACo requires two forward passes per decoding step. Unlike prompting baselines (in-context resampling, paraphrase prompting) that require n sequential decoding passes, diverse beam search (≥n beam expansions), or training-time methods, BACo's ~2× FLOPs overhead is a fixed multiplier independent of sample count. When the implementation runs both models simultaneously, the wall-clock time approaches 1× despite 2× total FLOPs. Though our paper focuses on the idea and prototype, further engineering-level optimizations are complementary and applicable when deploying at scale, e.g., speculative decoding [1], LoRA-aligned models, and KV cache sharing (discussed in §A.4).
>
> **[New Experiment] Wall-clock runtime comparison.** We report wall-clock runtime under the same hardware (A100 80GB), focusing on inference-time methods for fair comparison. All methods use the HuggingFace library.
>
> ***Results:***
>
> | Method | Time/Sample (s) | Tok/s (final output) | Peak Memory |
> |--------|-----------------|-------|-------------|
> | Aligned | 9.8±3.9 | 40.0 | 15.1 GiB |
> | Response Ensemble | 10.9±4.0 | 34.8 | 15.4 GiB |
> | Diverse Beam Search | 17.0±5.9 | 17.0 | 15.8 GiB |
> | Logits Ensemble | 25.5±0.5 | 20.1 | 30.1 GiB |
> | In-Context Resampling | 90.5±35.3 | 4.1 | 15.5 GiB |
> | Paraphrase Prompting | 117.9±41.1 | 3.3 | 15.1 GiB |
> | Back Translation | 120.5±65.6 | 3.8 | 15.1 GiB |
> | Mix-of-Agents | 224.9±123.2 | 3.8 | 15.6 GiB |
> | Multi-Agent Debate  | 239.0±81.5 | 3.5 | 15.6 GiB |
> | BACo | 22.9±4.5 | 19.2 | 29.2 GiB |
>
> [Table 1](https://gist.github.com/anonymous-rebuttal-anonymous/4850f1fcea39eed8321034677a808daf?permalink_comment_id=6067929#gistcomment-6067929): Wall-clock runtime comparison.
>
>
> ***Findings:*** BACo's current prototype is slower than single-model baselines, but is comparable to or cheaper than some other baselines while significantly outperforming them. The implementation is not yet engineered for speed, and the optimizations above (§A.4) can further improve it.
>
> ---
>
> **W2/Q2: Multi-turn Settings**
>
> **[New Experiment] MTBench evaluation.** We run BACo on MTBench to assess performance in multi-turn settings. The experiment setting follows §5.1 "Validation beyond open-ended evaluation metrics."
>
> ***Results:***
>
> | Method | Coverage | Dominance |
> |--------|----------|-----------|
> | Aligned | 0.320 | 27.2% |
> | BACo-P-Punc | **0.681** | **72.8%** |
>
> **Please refer to the [anonymous figure](https://gist.github.com/anonymous-rebuttal-anonymous/4850f1fcea39eed8321034677a808daf?permalink_comment_id=6065998#gistcomment-6065998) for trade-off plotting.**
>
>
>
> ***Findings:*** BACo achieves 0.681 Coverage and 72.8% Dominance, substantially outperforming the aligned baseline (0.320 and 27.2%) consistently across different diversity and quality metrics. Increasing temperature can fail to improve some diversity metrics.
>
>
> ---
>
> **W2 (continued): RL Applicability**
>
> We acknowledge RL compatibility as an interesting direction. BACo in this paper is primarily intended as an inference-time method for open-ended tasks. Applying BACo to RL (e.g., rollout) is worth exploring but beyond the scope of this paper.
>
>
> [1] Leviathan et al., "Fast inference from transformers via speculative decoding." ICML, 2023.

---

> > ### Author Rebuttal · Reviewer_hL6H · 2026-04-06
> >
> > My concerns have been addressed. I maintain my positive score.

---

> > > ### Author Response · Authors · 2026-04-06
> > >
> > > We thank Reviewer hL6H for the acknowledgement and the positive assessment.

---

### Decision · Program_Chairs · 2026-04-30

**Decision:**

Accept (regular)

**Comment:**

This paper introduces Base-Aligned Model Collaboration (BACo), a novel and effective method for enhancing the diversity and quality of generations from large language models. It is based on the observation that (i) Base LLMs are often strong in diversity but weak in quality, and (ii) Aligned (e.g., through Instruction-tuning) LLMs are strong in quality but weak in diversity. It designs a decoding method that routes between the base model (e.g. Llama3-8b) and the aligned model (llama-3-8b-instruct) to achieve better quality-diversity tradeoffs.

The reviewers were unanimously positive. The authors provided a thorough and convincing rebuttal that fully addressed all questions and pledged to incorporate these clarifications into the final version. Overall, this paper is recommended to be accepted.